# NEMO reshapes the α-Synuclein aggregate interface and acts as an autophagy adapter by co-condensation with p62

Nikolas Furthmann [1], Verian Bader [1,2], Lena Angersbach[1], Alina Blusch[3], Simran Goel[1], Ana Sánchez-Vicente[1], Laura J. Krause [1,4], Sarah A. Chaban [1], Prerna Grover [2], Victoria A. Trinkaus[5], Eva M. van Well[1], Maximilian Jaugstetter[6], Kristina Tschulik [4,6], Rune Busk Damgaard [7], Carsten Saft[3], Gisa Ellrichmann[3,20], Ralf Gold[3], Arend Koch[8], Benjamin Englert [8,21], Ana Westenberger[9], Christine Klein[9], Lisa Jungbluth [10,11], Carsten Sachse [10,11,12], Christian Behrends [13], Markus Glatzel [14], F. Ulrich Hartl [5,15], Ken Nakamura[16,17], Chadwick W. Christine [17,18], Eric J. Huang [17,19], Jörg Tatzelt[2,4] & Konstanze F. Winklhofer [1,4] ✉

NEMO is a ubiquitin-binding protein which regulates canonical NF-κB pathway activation in innate immune signaling, cell death regulation and host-pathogen interactions. Here we identify an NF-κB-independent function of NEMO in proteostasis regulation by promoting autophagosomal clearance of protein aggregates. NEMO-deficient cells accumulate misfolded proteins upon proteotoxic stress and are vulnerable to proteostasis challenges. Moreover, a patient with a mutation in the NEMO-encoding *IKBKG* gene resulting in defective binding of NEMO to linear ubiquitin chains, developed a widespread mixed brain proteinopathy, including α-synuclein, tau and TDP-43 pathology. NEMO amplifies linear ubiquitylation at α-synuclein aggregates and promotes the local concentration of p62 into foci. In vitro, NEMO lowers the threshold concentrations required for ubiquitin-dependent phase transition of p62. In summary, NEMO reshapes the aggregate surface for efficient autophagosomal clearance by providing a mobile phase at the aggregate interphase favoring co-condensation with p62.

Cellular proteostasis is maintained by an elaborate network of protein quality control components, such as molecular chaperones, the ubiquitylation machinery, the proteasome, and the autophagy-lysosome pathway, to coordinate the proper triage, subcellular quarantine, and refolding or disposal of damaged proteins. In aging, proteotoxic stress along with a decline in the fidelity of protein quality control favors the accumulation of misfolded proteins, challenging the integrity of the cellular proteome. This is particularly relevant to postmitotic cells, such as neurons. In fact, the age-dependent accumulation of misfolded proteins is a characteristic feature of neurodegenerative diseases. Cells are equipped with two proteolytic machineries, the ubiquitin-proteasome system and the autophagy-lysosomal system that remove misfolded proteins depending on their subcellular localization, their structure, oligomerization state and posttranslational modifications. Both degradation pathways are critically regulated by ubiquitin in cooperation with specific ubiquitin-binding proteins that target cargo to the proteasome or to selective autophagy[1–6].

The selective autophagic degradation of aggregated proteins, referred to as aggrephagy, is mediated by several cargo receptors, such as p62/SQSTM1, NBR1, and TAX1BP1[7–11]. p62 and NBR1 both bind

to ubiquitinated cargo by their UBA (ubiquitin-associated) domains and cooperate as hetero-oligomers in forming local clusters on the cargo[12–16]. NBR1 recruits TAXBP1, which drives autophagosome biogenesis by interacting with FIP200, a scaffold protein of the core autophagy machinery[16–18]. Tethering of the aggregated cargo to the growing autophagosomal membrane is mediated by LC3 proteins, which bind to LIR (LC3-interacting region) motifs present in p62, NBR1, and TAXBP1. Finally, the cargo is engulfed by a double membrane and degraded upon fusion of the autophagosome with lysosomes.

Ubiquitylation is a multifaceted posttranslational modification, regulating a plethora of cellular processes. Its complexity is based on different variables, such as the number of ubiquitin molecules attached to substrates, the mode of inter-ubiquitin linkage, the formation of homo- and heterotypic chains, and the posttranslational modification of ubiquitin itself, for example by phosphorylation, SUMOylation, or NEDDylation[19–22]. In conventional ubiquitylation, the C-terminal glycine of ubiquitin is linked to one of seven lysine residues of another ubiquitin molecule by an isopeptide bond. Alternatively, the C-terminal glycine can be linked to the N-terminal methionine of the acceptor ubiquitin, resulting in the formation of a peptide bond[23]. This type of head-to-tail linkage is called linear or M1-linked ubiquitylation. Linear ubiquitin chains are exclusively generated by the RING-in-between-RING (RBR) E3 ubiquitin ligase HOIP, the catalytic component of the linear ubiquitin chain assembly complex (LUBAC). Within this complex, HOIP interacts with HOIL-1 and SHARPIN, which bind to the UBA domain of HOIP by their ubiquitin-like (UBL) domains, thereby activating autoinhibited HOIP[24–33]. LUBAC has mostly been studied in the context of immune signaling, NF-κB activation, and cell death regulation[34–38]. In these paradigms, the regulatory component of the IKK (inhibitor of κB kinase) complex NEMO (nuclear factor-κB essential modulator), also called IKKγ, is an important player, serving both as an interactor of linear ubiquitin chains and a substrate of HOIP[39–41]. Binding of NEMO to M1-linked ubiquitin chains via its UBAN (ubiquitin-binding in ABIN and NEMO) domain and also ubiquitylation of NEMO upon activation of innate immune receptors, such as the TNF receptor 1, induces a conformational change in NEMO and activates the associated kinases IKKα and IKKβ[26,42,43]. The activated IKK kinases phosphorylate IκBα (inhibitor of κBα), which is then modified by K48-linked ubiquitin chains and degraded by the proteasome. Thereby, NF-κB heterodimers, typically p65 and p50, are liberated from their binding to IκBα and translocate to the nucleus to regulate the expression of NF-κB-dependent genes.

Amorphic or hypomorphic mutations in the NEMO-encoding *IKBKG* gene located on the X-chromosome are associated with *Incontinentia pigmenti* (IP), a condition that is usually lethal in male fetuses and thus almost exclusively occurs in female patients with mosaic X-chromosome inactivation[44–46]. IP is a rare multisystem disorder that primarily affects the skin, but can also involve other ectodermal tissues including teeth, hair, nails, eyes, and the central nervous system[47]. IP-linked mutations comprise rearrangements, nonsense, frameshift, splice site, or missense variants[45,46]. Although IP can be inherited in an X-linked dominant fashion, between 65 and 75% of cases occur sporadically due to de novo mutations[48]. The phenotypic heterogeneity of the clinical presentation is based on the fact that throughout the body, cells expressing the wildtype *IKBKG* allele co-exist with cells expressing the mutant *IKBKG* allele, hampering the analysis of patients' biosamples and the establishment of patient-derived cellular models. We identified a patient with IP who developed early onset, rapidly progressive neurodegeneration with a widespread mixed proteinopathy, including α-synuclein, tau, and TDP-43 aggregates. The pathogenic p.Q330* NEMO variant is compromised in NF-κB pathway activation, explaining the manifestation of IP, but in addition causes defective protein quality control. NEMO deficiency favors the accumulation of misfolded proteins and impairs the clearance of protein aggregates by autophagy. In this work, we reveal a critical function of NEMO in

proteostasis regulation by remodeling the aggregate interface and facilitating condensate formation of the autophagy cargo receptor p62.

## Results

### A pathogenic variant in the *IKBKG* gene encoding NEMO is associated with a widespread mixed proteinopathy and progressive neurodegeneration

A female patient, who was diagnosed with IP during her infancy developed parkinsonism (left-sided bradykinesia, hypophonia, slowed gait) at age 48 years. She noted initial benefit with treatment with carbidopa/levodopa, but her condition rapidly progressed, forcing her retirement from work as a school teacher at age 52 years. A dopamine transporter scan was markedly abnormal, showing near absence of tracer uptake within the caudate and putamen bilaterally. She then developed severe cognitive impairment and died at age 56 years from progressive neurodegeneration. Brain autopsy revealed a widespread mixed proteinopathy, including α-synuclein, tau, and TDP-43 aggregates with predominant α-synuclein pathology (Fig. 1a). There was no family history of neurodegenerative diseases and whole-genome sequencing did not identify pathogenic or likely pathogenic variants in genes associated with neurodegenerative disorders (Supplementary Fig. 1). Genetic testing revealed a mosaic c.988 C > T nonsense mutation in the *IKBKG* gene, replacing glutamine at position 330 by a premature stop codon (p.Gln330*). If translated, the resulting p.Q330* NEMO variant would be truncated and lack the C-terminal region encompassing the leucine zipper (LZ) and zinc finger (ZF) (Fig. 1b).

Based on our previous finding that NEMO is recruited to misfolded huntingtin with a polyglutamine expansion (Htt-polyQ)[49], we investigated whether an *IKBKG* gene mutation resulting in NEMO dysfunction is causally linked to neuronal protein aggregation observed in our patient. A functional characterization of NEMO p.Q330* (hereafter referred to as Q330X) in cellular models showed that this mutant is defective in NF-κB signaling, explaining the clinical presentation with IP. In contrast to wildtype (WT) NEMO, Q330X NEMO was not able to promote TNF-induced degradation of IκBα or NF-κB transcriptional activity (Supplementary Fig. 2a–d). We observed impaired NF-κB activation by Q330X NEMO also upon IL-1β receptor activation[50]. However, our previous study revealed that HOIP was able to decrease Htt-polyQ aggregates independently of NF-κB activation[49]. Therefore, we studied whether NEMO is a downstream effector of LUBAC-mediated protein quality control. First, we tested if endogenous NEMO is recruited to protein aggregates other than Htt-polyQ. Indeed, immunohistochemistry of brain slices from patients with Parkinson's disease (PD), Alzheimer's disease (AD) or frontotemporal dementia (FTD) provided evidence for both NEMO and M1-linked ubiquitin chains co-localizing at aggregates formed by α-synuclein, tau, or TDP-43 (Fig. 1c, d).

### NEMO deficiency promotes protein aggregation under proteotoxic stress

To analyze a possible role of NEMO in proteostasis regulation, we quantified the amount of protein aggregates induced by heat stress, proteasomal and lysosomal inhibition in wildtype and NEMO knockout (KO) embryonic fibroblasts (MEFs) by Proteostat®, a red fluorescent molecular rotor dye that binds to the cross-beta spine quaternary structure of aggregated proteins. In the absence of NEMO, protein aggregation was significantly increased in response to both heat stress and lysosomal inhibition, and a trend towards more aggregates was observed upon proteasomal inhibition (Fig. 2a, b). In addition, NEMO-deficient cells were more vulnerable to proteotoxic stress. Cell viability of NEMO KO MEFs was significantly decreased in response to heat stress, proteasomal or lysosomal inhibition compared to wildtype NEMO MEFs (Supplementary Fig. 3a–c). We then made use of a luciferase-based sensor of proteostasis capacity, the conformationally

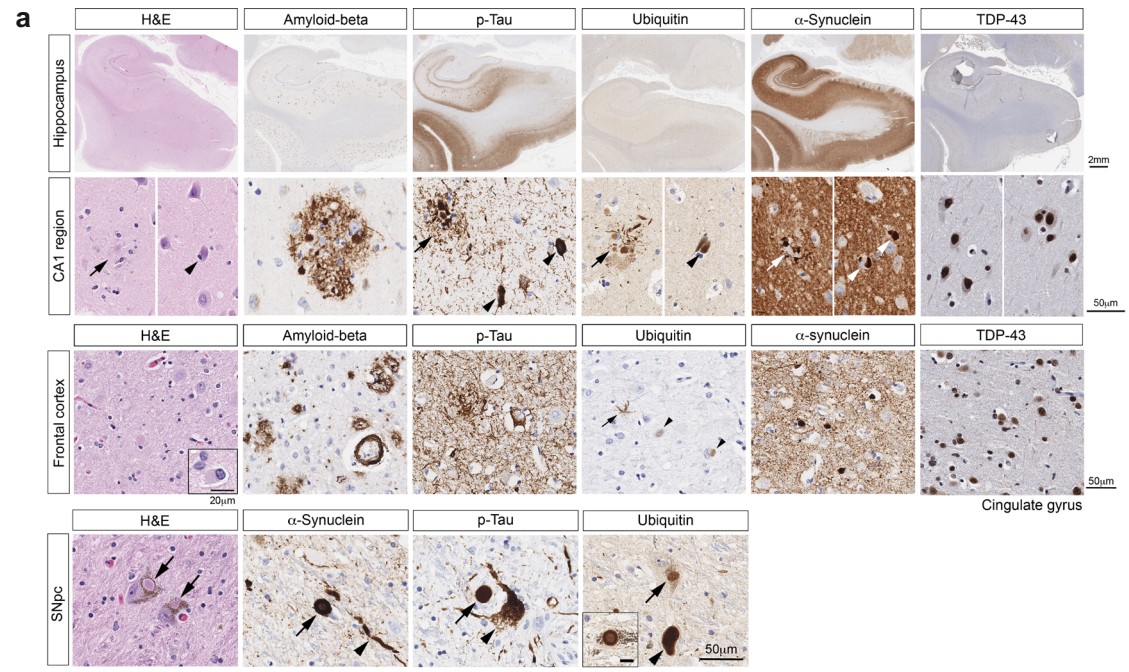

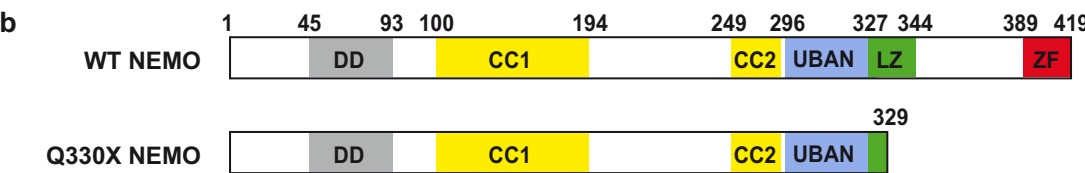

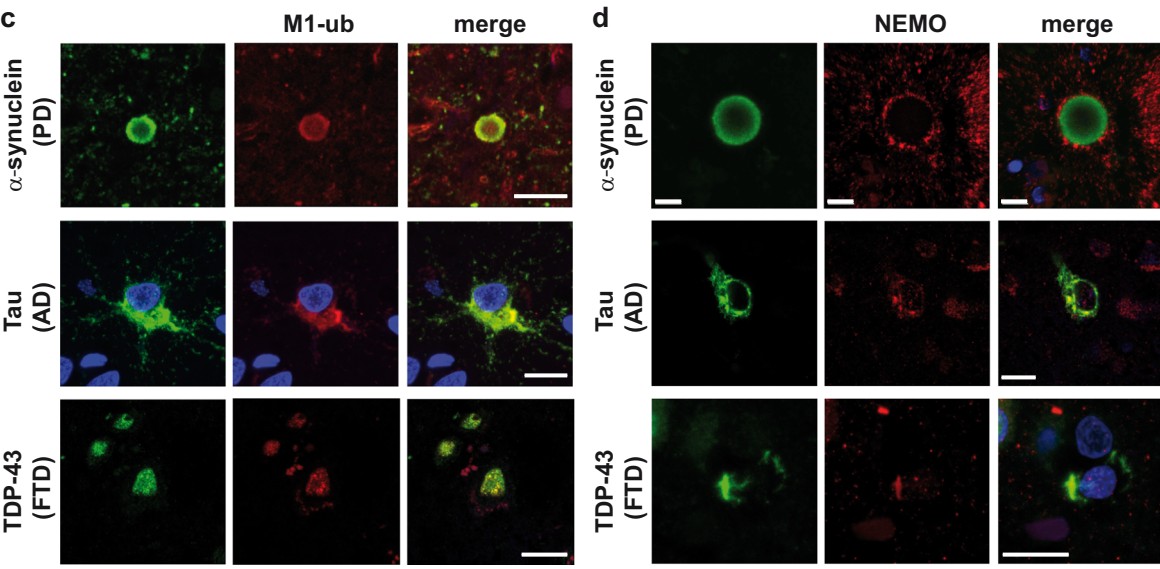

destabilized double mutant firefly luciferase FlucDM (R188Q, R261Q) fused to EGFP[51,52]. Imbalances in proteostasis induce misfolding and aggregation of this sensor, which can be monitored by the formation of FlucDM-EGFP-positive foci and a decrease in its luciferase activity. With the help of this sensor we compared the effects of wildtype and Q330X NEMO under basal and proteotoxic stress conditions in a NEMO-deficient background. NEMO KO MEFs transiently expressing

FlucDM-EGFP and either wildtype or Q330X NEMO were quantified for the abundance of EGFP-positive foci and for luciferase activity. Heat stress (43 °C, 20 min) increased FlucDM-EGFP foci formation and decreased its luciferase activity in NEMO KO MEFs (Fig. 2c, d). Expression of wildtype NEMO significantly reduced foci formation and increased luciferase activity of FlucDM-EGFP under both basal and heat stress conditions, whereas Q330X NEMO had no effect (Fig. 2c, d).

**Fig. 1 | NEMO is associated with pathological protein aggregates. a** Widespread mixed brain proteinopathy in a patient expressing mutant Q330X NEMO. Both low and high magnification images show the presence of aggregated proteins, such as α-synuclein, hyperphosphorylated tau, TDP-43, and amyloid beta in different brain regions. Many structures resembling Lewy body and Lewy neurites in pigmented neurons in the substantia nigra pars compacta (SNpc) are positive in immunostaining for α-synuclein, hyperphosphorylated tau, and ubiquitin. Scale bars, as indicated. **b** Domain structure of wildtype (WT) human NEMO and mutant Q330X

NEMO. DD dimerization domain, CC1 coiled-coil 1 domain, CC2 coiled-coil 2 domain, UBAN ubiquitin-binding in ABIN and NEMO, LZ leucine zipper, ZF zinc finger. **c, d** M1-linked ubiquitin and NEMO co-localize with α-synuclein, tau, and TDP-43 aggregates in human brain. Immunofluorescent stainings of cortical or midbrain sections from patients with Parkinson's disease (PD), Alzheimer´s disease (AD), or frontotemporal dementia (FTD). Brain sections were stained with antibodies against M1-ubiquitin (**c**), NEMO (**d**), and α-synuclein (PD), tau (AD), or TDP-43 (FTD); DAPI (blue). Scale bar, 10 μm.

Interestingly, NF-κB was not activated by heat stress, as shown by an NF-κB luciferase reporter assay and a p65 nuclear translocation assay (Fig. 2e, f). In conclusion, NEMO plays a role in proteostasis regulation that seems to be independent of NF-κB activation.

### Misfolded α-synuclein is decorated with M1-linked ubiquitin and NEMO

Since the NEMO mutant patient showed a predominant α-synuclein pathology, we employed cellular models of α-synuclein (aSyn) aggregation for mechanistic studies. Recombinant A53T aSyn preformed fibrils (PFFs) were added as seeds to SH-SY5Y cells stably expressing A53T aSyn fused to GFP (aSyn-GFP) to induce aggregation of aSyn-GFP[53–55]. Intracellular aSyn aggregates formed upon seeding showed characteristic features of pathologic aSyn, such as phosphorylation at serine 129 and insolubility in detergents, as determined by immunocytochemistry and immunoblotting, respectively (Supplementary Fig. 4a, b). In addition, we characterized the aSyn seeds by Thioflavin T fluorescence, dynamic light scattering, and liquid atomic force microscopy (Supplementary Fig. 5a–f).

Consistent with our observation that M1-linked ubiquitin occurs at Lewy bodies in human brain (Fig. 1c), linear ubiquitin chains co-localized with aSyn aggregates in SH-SY5Y cells after seeding (Fig. 3a). Moreover, seeding of primary cortical neurons induced misfolding and serine 129 phosphorylation of endogenous aSyn, which stained positive for M1-linked ubiquitin (Fig. 3b). The presence of linear ubiquitin chains at aSyn in seeded SH-SY5Y cells was confirmed biochemically by affinity purification of aSyn-GFP followed by immunoblotting using M1-ubiquitin-specific antibodies[56] (Fig. 3c). Moreover, endogenous NEMO was significantly enriched at seeded aSyn aggregates (Fig. 3d) and all three LUBAC components were recruited to aSyn aggregates (Fig. 3e). In contrast, the RBR E3 ubiquitin ligase ARIH1 (Ariadne homolog 1), used as a control, was not recruited to aSyn aggregates (Fig. 3e).

### The Q330X NEMO mutant does not bind to M1-linked ubiquitin chains and is not ubiquitylated by HOIP

To gain further insight into the function of NEMO in proteostasis regulation, we compared the effect of wildtype NEMO and Q330X NEMO on aSyn aggregates in the SH-SY5Y seeding model. We first tested for NEMO recruitment and observed that Q330X NEMO did not efficiently co-localize with aSyn aggregates, whereas wildtype NEMO was strongly enriched at aSyn aggregates (Fig. 4a). Defective recruitment of Q330X NEMO to protein aggregates was confirmed in SH-SY5Y cells with Htt-polyQ (Htt-Q97) aggregates, indicating that this phenomenon is not limited to aSyn aggregates (Fig. 4b). We also performed filter retardation assays using the SDS-insoluble fraction of Htt-Q97-expressing NEMO KO MEFs reconstituted with either wildtype, Q330X or K285R/K309R NEMO. The K285R/K309R NEMO mutant lacks two critical lysine residues required for NEMO linear ubiquitylation[57]. In contrast to wildtype NEMO, Q330X or K285R/K309R NEMO were less efficiently retained by the membrane together with Htt-Q97, suggesting decreased abundance of these mutants at Htt-Q97 aggregates (Fig. 4c).

We previously found that NEMO is recruited to Htt-polyQ aggregates by binding to linear ubiquitin chains generated by HOIP[49]. Therefore, we compared the capacity of wildtype NEMO and Q330X

NEMO to bind linear ubiquitin chains. Lysates of cells expressing either wildtype or Q330X HA-tagged NEMO were incubated with recombinant tetra-M1-ubiquitin (4×M1-ub), then NEMO was affinity-purified via its HA tag and immunoblotted using M1-ubiquitin-specific antibodies. In contrast to wildtype NEMO, Q330X NEMO did not bind to tetra-M1-ubiquitin, although the UBAN domain (residues 296–327) is not affected by the C-terminal deletion (Fig. 4d). Moreover, Q330X NEMO was not modified by linear ubiquitin chains, although the key ubiquitin acceptor lysine residues (K285 and K309) are present in this mutant[57]. We induced M1-ubiquitination of HA-tagged NEMO by either treating the cells with TNF or overexpressing the LUBAC components HOIP, HOIL-1L, and SHARPIN (Fig. 4e). The cells were lysed under denaturing conditions, NEMO was immunoprecipitated via its HA tag and analyzed by immunoblotting using M1-ubiquitin-specific antibodies. Whereas wildtype NEMO was M1-ubiquitylated in both conditions, Q330X NEMO was not even ubiquitylated upon the overexpression of LUBAC (Fig. 4e). Next, we analyzed the interaction of wildtype NEMO and Q330X NEMO with endogenous HOIP by co-immunoprecipitation experiments. The NZF1 domain of HOIP interacts with the coiled-coil 2 (CC2) region upstream the UBAN domain of NEMO[57–59]. HOIP did not co-purify with Q330X NEMO, which helps to explain why Q330X NEMO is not ubiquitylated (Fig. 4f).

Since NEMO not only binds to M1-linked polyubiquitin but also is covalently modified by HOIP, we followed up the hypothesis that NEMO increases M1-ubiquitylation at aSyn aggregates. We generated NEMO KO SH-SY5Y cells by CRISPR/Cas9 and induced aggregation of transiently expressed aSyn-GFP by adding aSyn seeds. NEMO KO SH-SY5Y cells were reconstituted with either wildtype NEMO or Q330X NEMO and the colocalization of M1-linked ubiquitin, NEMO, and aSyn aggregates was quantified by the Pearson coefficient. Colocalization of M1-linked ubiquitin and aSyn was strongly reduced when Q330X NEMO was expressed in comparison to wildtype NEMO (Fig. 4g, h, Supplementary Fig. 6a), suggesting that NEMO amplifies M1-ubiquitylation at aSyn aggregates. A disadvantage of the aSyn seeding model regarding biochemical studies is the low aggregation efficiency due to the fact that PFFs are taken up only by a subfraction of cells. To add more evidence for the role of NEMO in increasing the abundance of M1-linked ubiquitin at aggregates, we therefore used cells expressing Htt-Q97, resulting in aggregate formation in all cells transfected with Htt-Q97. HEK293T cells expressing Htt-Q97 or Htt-Q97 together with either wildtype NEMO, Q330X NEMO, or D311N NEMO, a mutant that is defective in binding to M1-linked ubiquitin[60,61], were lysed 72 h after transfection under denaturing conditions. The SDS-insoluble pellets containing the Htt-Q97 aggregates were dissolved in formic acid and analyzed by immunoblotting. Wildtype NEMO but neither Q330X nor D311N NEMO increased the M1-linked ubiquitin-specific signal, supporting the M1-ubiquitin-amplifying function of NEMO at protein aggregates (Fig. 4i).

### A local NF-κB signaling platform is assembled at aSyn aggregates which does not promote a functional response

It has been shown recently that LUBAC is recruited to intracellular bacteria, such as *Salmonella enterica*, which can escape vacuoles and invade the cytosol[62–64]. Modification of the bacterial surface with linear ubiquitin chains by HOIP induces anti-bacterial autophagy (xenophagy) and recruits NEMO for local activation of NF-κB. We therefore

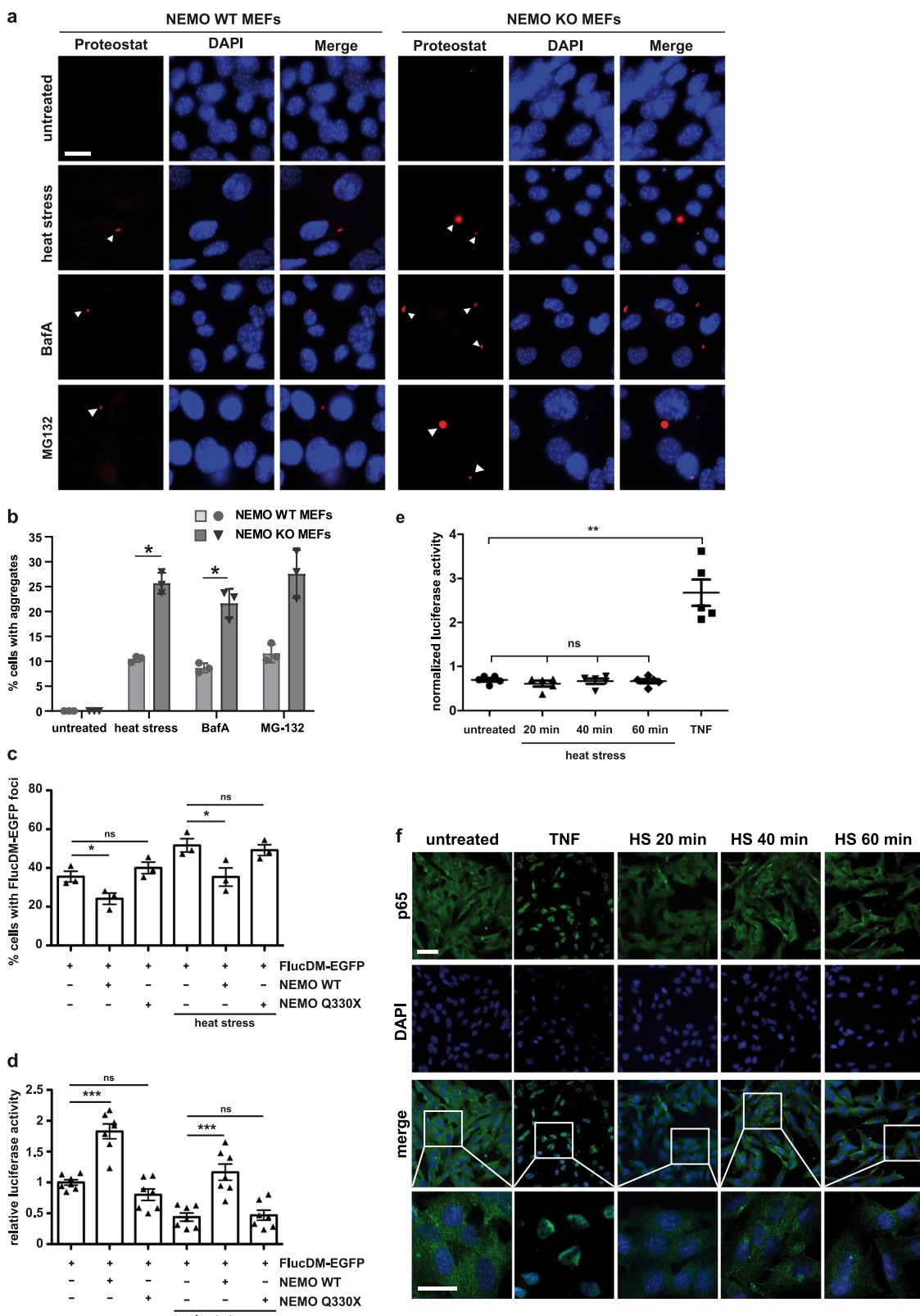

tested whether an NF-κB signaling platform is also assembled at aSyn aggregates. Indeed, endogenous phospho-IKKα/β (p-IKKα/β), p65, and phospho-p65 (p-p65) were strongly enriched at aSyn aggregates (Fig. 5a). However, we did not observe nuclear translocation of the NF-κB subunit p65 on day 1, 2, or 3 after seeding (Fig. 5b, c). To test whether NF-κB signaling is compromised in the presence of aSyn aggregates, we treated aSyn-expressing cells with TNF on day 1, 2 and 3

after seeding. Whereas almost 100% of non-seeded cells showed nuclear translocation of p65 in response to TNF treatment, p65 translocation was significantly reduced upon seeding with only about 50% of cells positive for nuclear p65 on day 3 after seeding (Fig. 5b, c). Super-resolution structured illumination fluorescence microscopy (SR-SIM) revealed that p65 is trapped at aSyn aggregates upon TNF stimulation, whereas in cells without aSyn aggregates, TNF-induced

**Fig. 2 | NEMO protects from proteotoxic stress. a b** NEMO-deficient cells are prone to protein aggregation under proteotoxic stress**. a** NEMO wildtype (WT) and knockout (KO) embryonic fibroblasts (MEFs) were heat stressed (42 °C, 1 h) or treated with the proteasomal inhibitor MG-132 (0.5 μM, 48 h) or the lysosomal inhibitor Bafilomycin A1 (BafA, 100 nM, 48 h) and then stained by Proteostat® to detect protein aggregates. Scale bar: 20 μm. **b** Cells positive for aggregates were quantified. All data are displayed as mean ± SD based on 3 independent experiments, analyzed by two-way ANOVA followed by Šídák's multiple comparisons test. At least 150 cells were assessed per condition. Heat stress: *$p = 0.0283$, BafA: *$p = 0.0414$. **c, d** Wildtype NEMO but not Q330X NEMO decreases misfolding of the folding sensor FlucDM-EGFP-luciferase. NEMO KO MEFs transiently expressing FlucDM-EGFP-luciferase and either wildtype (WT) NEMO or Q330X NEMO were subjected to a heat stress (HS, 43 °C, 20 min) 48 h after transfection or left untreated. **c** The cells were then analyzed by immunocytochemistry and fluorescence microscopy. Shown is the fraction of NEMO-expressing cells with EGFP-

positive foci. Data represent mean ± SEM based on 3 independent experiments. At least 900 transfected cells have been analyzed per condition. Statistics: one-tailed Mann–Whitney $U$-tests *$p \le 0.05$. **d** In parallel, luciferase activity of control and heat stressed cells were analyzed luminometrically. Data represent mean ± SEM based on 7 independent experiments. Statistics: One-way ANOVA with Bonferroni's multiple comparison posthoc test; ***$p \le 0.001$. **e, f** Transient heat stress does not activate NF-κB signaling. **e** HEK293T cells transiently expressing an NF-κB-luciferase reporter construct were heat stressed for 20, 40, or 60 min (42 °C) and 8 h later luciferase activity was quantified. As a positive control, one set of cells was treated with TNF (10 ng/ml, 8 h). Data are shown as normalized mean ± SD based on 5 independent biological replicates. **$p \le 0.01$. **f** SH-SY5Y cells were heat stressed (42 °C) for the indicated time and then nuclear translocation of the NF-κB subunit p65 was analyzed by immunocytochemistry and fluorescence microscopy using antibodies against p65. As a positive control, one set of cells was treated with TNF (25 ng/ml, 15 min). Scale bar, 10 μm (overview) and 10 μm (inset).

efficient nuclear translocation of p65 (Fig. 5c). Moreover, TNF-induced nuclear translocation was also impaired in SH-SY5Y cells expressing Htt-Q97, suggesting that pathogenic protein aggregates interfere with NF-κB signaling (Fig. 5d). These results indicated that although NF-κB is locally activated at aggregates, functional NF-κB signaling is impaired most probably through sequestration of p65 and possibly other NF-κB pathway components at the aggregates.

## NEMO promotes autophagosomal degradation of α-synuclein in a p62-dependent manner

Our previous data indicated that NEMO is recruited to aSyn aggregates along with other NF-κB signaling components without inducing a functional NF-κB response. Thus, NEMO seems to have an NF-κB-independent role in proteostasis regulation. Along this line, different protein aggregates accumulated in the brain of the Q330X NEMO patient, and wildtype NEMO but not Q330X NEMO reduced the fraction of cells with FlucDM-EGFP aggregates. LUBAC promotes the autophagic clearance of cytosol-invading bacteria[62–64], we therefore wondered whether linear ubiquitin chains can influence the degradation of protein aggregates by autophagy.

Intracellular aSyn can be degraded by both the proteasome and lysosomes depending on its posttranslational modifications, conformational state, and subcellular localization[65–67]. To test whether linear ubiquitylation promotes degradation of aSyn aggregates via autophagy, we transiently expressed NEMO or HOIP in SH-SY5Y cells and quantified the fraction of cells with aSyn aggregates 48 h after seeding. Both wildtype NEMO and wildtype HOIP, but neither Q330X NEMO nor catalytically inactive C885A HOIP, decreased the number of cells with aSyn aggregates (Fig. 6a, b). Notably, inhibition of lysosomal degradation by bafilomycin A1 abolished the ability of NEMO and HOIP to reduce the number of cells with aSyn aggregates (Fig. 6a, b). We also tested the D311N NEMO mutant, compromised in binding to M1-linked ubiquitin[60,61]. Similarly to Q330X NEMO, M1- and K63-ubiquitylation of D311N NEMO was impaired (Supplementary Fig. 6b). Also, it was not recruited to aSyn aggregates (Supplementary Fig. 6c) and did not reduce aSyn aggregates (Fig. 6a). To validate our observations for endogenous NEMO and HOIP, we downregulated the expression of NEMO or HOIP by RNA interference in the aSyn seeding model. Silencing of NEMO or HOIP increased the number of cells with aSyn aggregates to a similar extent (Fig. 6c), corroborating our results with NEMO or HOIP overexpression.

Since NEMO and HOIP apparently promote the degradation of misfolded aSyn depending on lysosomal function, we aimed at uncovering the underlying mechanism. Ubiquitylation is decoded and translated into cellular effects by specific ubiquitin-binding proteins. We reasoned that p62/SQSTM1 might be a promising candidate to test in our paradigm, based on its key role in targeting protein aggregates for selective autophagy[5,12,68–70]. The ubiquitin-binding UBA domain of p62, which is required to shuttle cargo to the autophagic machinery,

binds to M1-linked ubiquitin with the strongest affinity compared to other ubiquitin linkages[71]. Moreover, p62 has been reported to interact with NEMO[72–74] and to co-localize with Lewy bodies[75,76]. We confirmed that endogenous p62 binds to aSyn aggregates in our seeding model (Fig. 6d) and that the recruitment of p62 to aSyn aggregates depends on its UBA domain (Supplementary Fig. 6d). Co-immunoprecipitation experiments using cell lysates revealed that in contrast to wildtype NEMO, the Q330X NEMO mutant does not interact with endogenous p62 (Fig. 6e).

To test for a role of p62 in mediating effects downstream of linear ubiquitylation, we analyzed aSyn aggregates in p62-deficient MEFs (p62 KO MEFs) expressing aSyn-GFP. Two days after seeding, about 75% of aSyn-GFP-expressing p62 KO MEFs displayed aSyn aggregates. Restoring p62 expression in p62 KO MEFs decreased the fraction of aggregate-positive cells to about 55%, similarly to the extent of aSyn aggregation observed in wildtype SH-SY5Y cells. Notably, the rescue effect of p62 was dependent on its UBA domain, since expression of p62-ΔUBA had no effect on the number of cells with aSyn aggregates in p62 KO MEFs (Fig. 6f). Increased expression of NEMO or HOIP in p62-deficient cells revealed a p62-dependent effect of both NEMO and HOIP, seen after reconstituting p62 KO MEFs with wildtype p62 in comparison to p62-ΔUBA (Fig. 6f). In addition, we observed a minor p62-independent effect in reducing aSyn aggregates, most probably mediated by other ubiquitin-binding autophagy receptors. Notably, the p62-dependent effect of NEMO was blocked by bafilomycin A1, confirming that p62 decreased aSyn aggregates downstream of NEMO by autophagosomal clearance (Fig. 6g).

Next, we wondered whether defective p62-dependent autophagosomal degradation of aSyn might explain the accumulation of aSyn aggregates in the Q330X NEMO patient. We analyzed brain sections from the Q330X NEMO patient by immunohistochemistry and found that colocalization of p62 with aSyn-positive aggregates was significantly reduced in comparison to patients suffering from other α-synucleinopathies, such as Dementia with Lewy Bodies (DLB) (Fig. 7a, b). Of note, p62 signal intensity was strongly increased in the Q330X NEMO patient's brain, possibly reflecting a compensatory upregulation of p62 expression (Fig. 7c). Encouraged by these findings, we wondered whether an influence of NEMO on p62 recruitment to aSyn aggregates is also evident in cellular models. We induced aggregation of transiently expressed aSyn-GFP in wildtype and NEMO KO SH-SY5Y cells by adding aSyn seeds and visualized endogenous p62 by SR-SIM, which revealed a striking difference in the pattern of p62 localization at the aggregates. Whereas p62 mostly formed foci at the aggregate surface in the presence of NEMO, it showed a more diffuse distribution around the aggregates in the absence of NEMO (Fig. 7d). The same NEMO-dependent differences in the localization pattern at aSyn aggregates were observed for the autophagy receptor NBR1 (Fig. 7e), which cooperates with p62 in forming local clusters on protein aggregates[16]. Ubiquitin-dependent p62 condensation is required for efficient

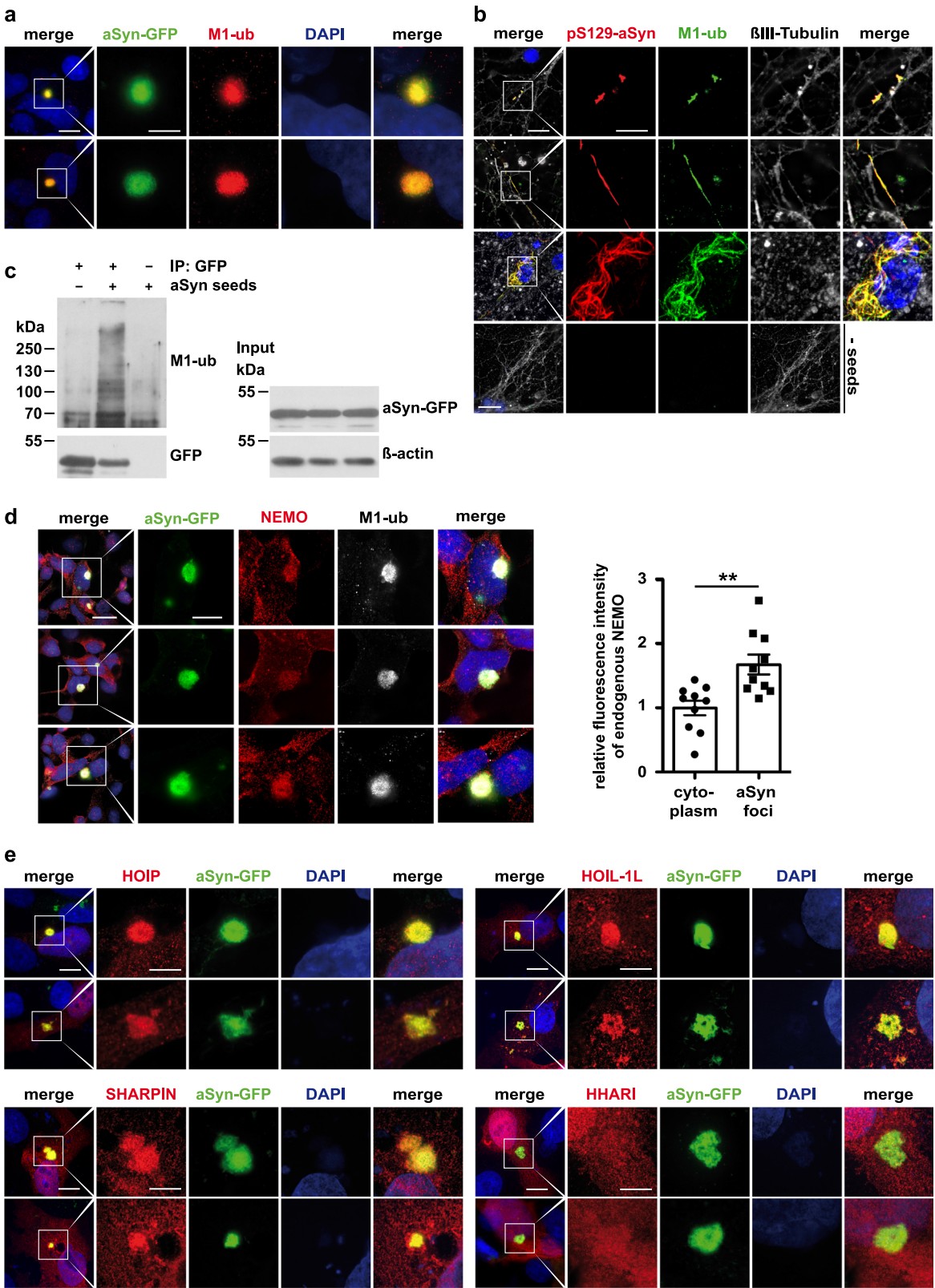

autophagic clearance of cargo[14–16,77,78], we therefore tested whether the non-condensed localization of p62 and NBR1 at aggregates has consequences on subsequent steps of the autophagic process. Indeed, the abundance of both LC3 and LAMP2 at aSyn aggregates was significantly decreased in NEMO KO cells (Fig. 7f, g). Thus, in the absence of NEMO, p62 and NBR1 are impaired in clustering at aggregated cargo, which compromises recruitment of the autophagic machinery.

## NEMO promotes p62 condensate formation at aggregates by lowering the threshold for ubiquitin-induced phase transition

We and others recently observed that NEMO undergoes phase separation upon binding to linear ubiquitin chains[50,79]. Since Q330X NEMO is impaired in binding to M1-linked ubiquitin, it does neither phase-separate in vitro nor form condensates in cells[50]. We therefore speculated that NEMO by interacting with M1-linked ubiquitin chains

**Fig. 3 | NEMO and LUBAC components are recruited to aSyn aggregates. a** M1-linked ubiquitin is enriched at aSyn aggregates formed in the cellular aSyn seeding model. SH-SY5Y cells stably expressing aSyn A53T-GFP were treated with aSyn A53T seeds, fixed 72 h after seeding, and analyzed by immunocytochemistry and fluorescence SR-SIM using M1-ubiquitin-specific antibodies. Scale bar, 10 μm (overview) and 5 μm (inset). **b** M1-linked ubiquitin colocalizes with pS129-aSyn-positive neurites in primary neurons. Primary cortical neurons were treated with aSyn A53T seeds at day 5 in vitro to induce aggregation of endogenous aSyn, fixed 7 days (rows 1 and 2) or 10 days (rows 3 and 4) after seeding, and analyzed by immunocytochemistry and fluorescence SR-SIM using antibodies against pS129-aSyn, M1-linked ubiquitin, and βIII-Tubulin. Scale bar, rows 1, 2, and 3: 10 μm (overview) and 5 μm (inset), row 4: 20 μm. **c** M1-linked ubiquitin co-immunoprecipitates with aSyn. SH-SY5Y cells stably expressing A53T-GFP aSyn were treated with aSyn A53T seeds for 72 h, lysed in 1% Triton X-100 in PBS, and aSyn-GFP was immunoprecipitated using GFP-trap beads. An immunoprecipitation with anti-HA beads was used to control for nonspecific binding. The pellet was analyzed by immunoblotting for M1-linked ubiquitin and GFP. The input was immunoblotted against aSyn and β-actin. **d** Endogenous NEMO is enriched at aSyn aggregates. SH-SY5Y cells stably expressing aSyn A53T-GFP were treated as described in a and analyzed by immunocytochemistry and fluorescence SR-SIM using antibodies against NEMO and M1-linked ubiquitin. Scale bar, 20 μm (overview) and 10 μm (inset). For quantification of the relative NEMO fluorescent signals, the NEMO-specific signal at aSyn-GFP aggregates was compared to a cytoplasmic area outside aSyn aggregates. The mean intensity within the cytoplasm was set to 1. Data represent mean ± SEM (n = 10). Statistics: Two-tailed t-test, **p = 0.0026. **e** LUBAC components are recruited to aSyn aggregates. SH-SY5Y cells stably expressing α-Synuclein A53T-GFP were transiently transfected with plasmids encoding either HA-HOIP, HA-HOIL-1 L, or HA-SHARPIN, or HA-HHARI as a control. One day after transfection, the cells were treated with aSyn A53T seeds, fixed 48 h after seeding, and analyzed by immunocytochemistry and fluorescence SR-SIM using antibodies against the HA-tag. Scale bar, 10 μm (overview) and 5 μm (inset).

at protein aggregates may prime the aggregate surface for efficient p62 condensate formation. Fluorescence recovery after photobleaching (FRAP) of Halo-tagged NEMO expressed in NEMO KO cells indicated that NEMO forms a mobile phase at aSyn aggregates, whereas the aSyn aggregates show a non-dynamic behavior (Fig. 8a, b). We also observed by SR-SIM microscopy that endogenous p62, NEMO, and M1-linked ubiquitin co-localize in condensates at aSyn aggregates (Fig. 8c). Next, we tested a possible role of M1-linked ubiquitin in mediating the interaction between NEMO and p62 by using recombinantly expressed proteins. Wildtype or Q330X NEMO fused to MBP (maltose-binding protein) was mixed with mCherry-p62 in the presence or absence of M1-linked tetra-ubiquitin (4×M1-ub). NEMO was immunoprecipitated by MBP-specific antibodies and immunoblotted for p62. In the absence of 4×M1-ub, the p62 signal upon co-purification with NEMO was minimally increased over background, but strongly increased in the presence of 4×M1-ub (Fig. 8d). Similarly to the co-immunoprecipitation experiments in cellular lysates, p62 did not co-purify with Q330X NEMO, confirming that defective binding of linear ubiquitin to Q330X NEMO also affects its interaction with p62 (Fig. 8d).

Our SR-SIM images indicated a foci-like staining also for NEMO and M1-linked ubiquitin at aggregates (Fig. 8c). This observation prompted us to study a possible effect of NEMO on ubiquitin-induced p62 condensation in vitro. We first tested whether p62 and NEMO can co-condensate. Recombinant mCherry-p62 and NEMO-GFP were mixed with or without recombinant tetra- or octa-M1-linked ubiquitin. Laser scanning microscopy revealed that co-condensation of p62 and NEMO occurred in presence of tetra- or octa-M1-linked ubiquitin, but not in the absence of M1-linked ubiquitin (Fig. 8e). We then studied p62 condensate formation dependent on the concentration of p62 and tetra- or octa-M1-linked ubiquitin (from 0.5 to 10 μM each) in the presence and absence of NEMO. As illustrated by phase diagrams, NEMO shifted p62 phase transition to the lowest concentration of both p62 and tetra- or octa-M1-ubiquitin (Fig. 8f). Thus, by co-condensation with p62 and M1-linked ubiquitin, NEMO facilitates the local concentration of p62. These results provide a mechanistic explanation for the impaired autophagic clearance of protein aggregates in the absence of functional NEMO observed in our cellular models and the Q330X NEMO patient brain samples.

## Discussion

Here we demonstrate that NEMO has an NF-κB-independent role in maintaining cellular proteostasis by acting as an autophagy adapter protein. NEMO-deficient cells accumulate misfolded proteins upon proteotoxic stress and are hypersensitive to proteostasis dysregulation. The crucial role of NEMO in maintaining proteostasis was confirmed by the neuropathological alterations found in the Q330X NEMO mutant patient, who shows a progressive widespread mixed brain proteinopathy with predominant aSyn pathology. Studying this NEMO variant revealed that LUBAC-mediated formation of linear ubiquitin chains and binding of NEMO to linear ubiquitin chains are required to promote autophagosomal degradation of misfolded aSyn through the selective autophagy receptor p62. We previously reported that wild-type NEMO in contrast to Q330X NEMO has the propensity to form phase-separated condensates upon binding to linear ubiquitin chains[50]. Here we show that through this propensity NEMO facilitates p62-dependent aggrephagy. Condensate formation with ubiquitinated cargo is a crucial event in p62-mediated autophagy, since it contributes to the recruitment of the autophagic machinery[14–16,77,78]. Whereas the mechanism of autophagy initiation by p62 has been studied in great detail, little is known about processes at the aggregate interface required to locally concentrate and rearrange p62. Our study identified NEMO as a major player in priming the aggregate interphase for p62 condensation. At least two NEMO-dependent processes seem to be relevant in this context. First, NEMO amplifies linear ubiquitination at protein aggregates. This is accomplished by both binding to M1-linked ubiquitin via its UBAN domain and being M1-ubiquitylated by HOIP. HOIP is recruited to protein aggregates by VCP/p97 and can generate free, unanchored M1-linked polyubiquitin[49] or assemble M1-linked ubiquitin chains on pre-existing K63-linked chains[80], suggesting that the aggregated protein not necessarily needs to be modified directly by HOIP. Moreover, NEMO is a HOIP substrate available at protein aggregates. Accordingly, we found M1-linked ubiquitin and NEMO at various protein aggregates, including aSyn, Htt-polyQ, tau, and TDP-43. Second, binding of NEMO to M1-linked ubiquitin generates a mobile phase-separated aggregate surface, facilitating the local concentration of p62 by co-condensation. Indeed, liquidity at the cargo surface seems to be an important prerequisite for efficient selective autophagy[81–85].

The Q330X NEMO mutant lacks the C-terminal region including part of the LZ and the ZF domain. The region between the CC2 (coiled-coil 2) and LZ domain includes the UBAN domain and forms elongated parallel coiled-coil dimers[59]. This region is required for binding to polyubiquitin, dimerization and interaction with HOIP[57–59]. The UBAN domain binds M1-linked polyubiquitin chains 100-fold stronger compared to K63-linked chains, whereas the C-terminal ZF binds K63-linked polyubiquitin chains with high affinity[61,86–90]. The ZF has also been reported to mediate binding to IκBα[91] and to the ubiquitin chain-editing enzyme A20, a negative regulator of NF-κB activation[92]. Although the UBAN domain is present in the Q330X NEMO mutant, it cannot bind to M1-linked ubiquitin, most likely caused by conformational alterations or impaired dimer formation of Q330X NEMO.

The UBAN domain is an M1-ubiquitin-binding domain present in NEMO, Optineurin, and ABIN-1-3 (A20-binding inhibitors of NF-κB)[87]. Whereas these UBAN proteins show differences in their ubiquitin-binding properties and their biological functions[93], they share a link to autophagy. Optineurin is a well-characterized autophagy receptor in

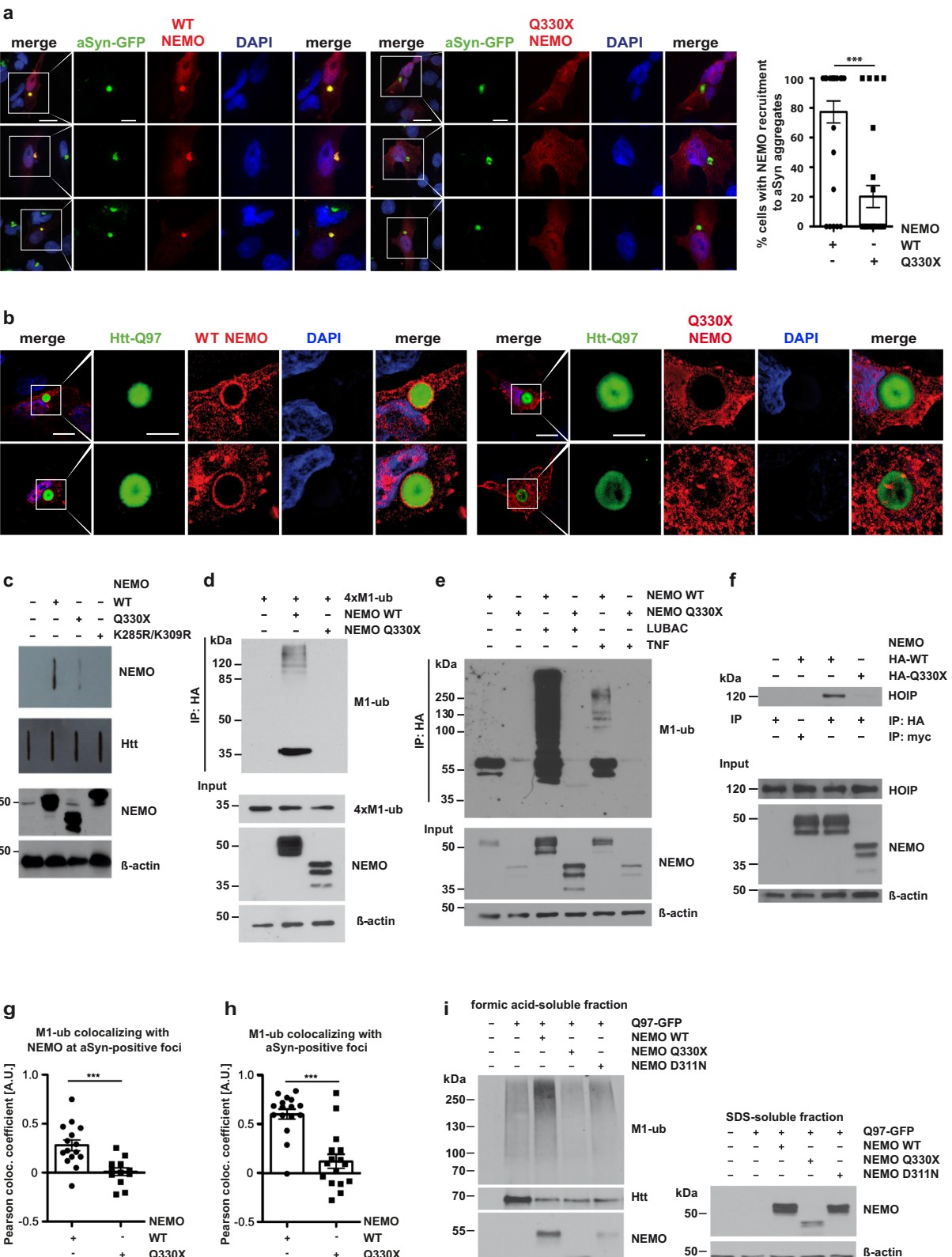

mitophagy, xenophagy and aggrephagy[8,11], and ABIN-1 has recently been linked to mitophagy[94]. In contrast to Optineurin and ABIN-1, NEMO has no LIR domain to directly interact with LC3. It rather functions as an indirect autophagy adapter protein by interacting with p62 in an M1-ubiquitin-dependent manner. Interestingly, the *Drosophila melanogaster* NEMO homolog Kenny has a LIR motif that interacts with Atg8/LC3 and promotes the autophagic degradation of the IKK

complex in order to prevent constitutive production of antimicrobial peptides against commensal microbiota[95]. According to a mathematical model proposed by Tusco et al., host-pathogen co-evolution could have been the driving force for the loss of the LIR motif in mammalian NEMO[95].

LUBAC-mediated quality control of cellular protein aggregates shares some similarities with its role in anti-bacterial autophagy.

**Fig. 4 | In contrast to Q330X NEMO, WT NEMO is recruited to pathological protein aggregates and increases the abundance of M1-linked ubiquitin. a** In contrast to WT NEMO, Q330X NEMO is not present at aSyn aggregates. SH-SY5Y cells stably expressing aSyn A53T-GFP were transiently transfected with either HA-tagged WT NEMO (left panel) or HA-tagged Q330X NEMO (right panel). After 24 h, the cells were treated with aSyn A53T seeds, fixed 48 h after seeding, and analyzed by immunocytochemistry and fluorescence SR-SIM using an antibody against the HA-tag. Scale bar, 20 μm (overview) and 10 μm (inset). The graph displays the percentage of cells showing colocalization of WT NEMO or Q330X NEMO with aSyn aggregates per field of view. Data are shown as mean ± SEM based on 8 technical replicates within 2 biological replicates. At least 41 cells per condition were quantified. Statistics: non-parametric Mann–Whitney U-test. ***$p \leq 0.001$. **b** Q330X NEMO is not recruited to Htt-Q97-GFP aggregates. SH-SY5Y cells were transiently transfected with Htt-Q97-GFP and either HA-tagged WT NEMO (left panel) or HA-tagged Q330X NEMO (right panel). Cells were fixed after 72 h and analyzed by immunocytochemistry and fluorescence SR-SIM using antibodies against the HA-tag. Scale bar, 10 μm (overview) and 5 μm (inset). **c** Q330X NEMO is not recruited to insoluble Htt-Q97-GFP aggregates. NEMO KO MEFs were transiently transfected with Htt-Q97-GFP and either WT HA-NEMO, Q330X HA-NEMO or K285R K309R HA-NEMO. After 72 h, the cells were lysed and the SDS-insoluble fractions were analyzed by a filter retardation assay. The cellulose acetate membranes were immunoblotted for Htt and NEMO (using an HA antibody). The input was immunoblotted using antibodies against HA and β-actin. **d** Q330X NEMO does not bind to M1-linked ubiquitin. HEK293T cells were transiently transfected with either WT HA-NEMO or Q330X HA-NEMO. After 24 h, the cells were lysed and lysates were incubated with recombinant M1-linked tetra-ubiquitin (4×M1-ub) for 2 h at 4 °C. NEMO was immunoprecipitated using anti-HA beads. Immunoprecipitated NEMO was immunoblotted using antibodies against M1-linked ubiquitin. The input was immunoblotted for M1-linked ubiquitin, NEMO and β-actin. **e** Q330X NEMO is not M1-ubiquitinated upon TNF treatment or increased LUBAC expression. HEK293T cells

were transiently transfected with WT HA-NEMO or Q330X HA-NEMO ± LUBAC (HOIP + HOIL-1L + SHARPIN) as indicated. After 24 h, one set of cells was treated with TNF (25 ng/ml, 30 min) to stimulate linear ubiquitination. After cell lysis under denaturing conditions, NEMO was immunoprecipitated using anti-HA beads. Immunoprecipitated NEMO was immunoblotted using antibodies against M1-linked ubiquitin. The input was immunoblotted for NEMO and β-actin. **f** In contrast to WT NEMO, Q330X NEMO does not co-immunoprecipitate with endogenous HOIP. HEK293T cells were transiently transfected with either WT HA-NEMO or Q330X HA-NEMO. After cell lysis, NEMO was immunoprecipitated using anti-HA beads. Anti-c-myc beads were used to control for nonspecific binding. Immunoprecipitated NEMO was immunoblotted for endogenous HOIP. The input was immunoblotted for HOIP, NEMO, and β-actin. **g, h** M1-ubiquitin chains at aSyn aggregates are increased by WT but not Q330X NEMO. CRISPR/Cas9 NEMO KO SH-SY5Y cells were transiently transfected with aSyn A53T-GFP and either WT NEMO or Q330X NEMO. One day after transfection, the cells were treated with aSyn A53T seeds, and fixed 48 h after seeding and analyzed by immunohistochemistry and fluorescence SR-SIM using antibodies against aSyn, M1-ubiquitin and NEMO. Colocalization of M1-ubiquitin, NEMO and aSyn aggregates (**g**) or M1-ubiquitin and aSyn aggregates (**h**) was quantified using the Pearson colocalization coefficient. **g** Data are displayed as mean ± SEM, $n = 16$ individual cells. Statistics: two-tailed Mann–Whitney U-test. ***$p = 0.0010$. **h** Data are displayed as mean ± SEM. $n = 18, 23$ individual cells. Statistics: two-tailed student's t-test. ***$p = 0.0003$. **i** NEMO Q330X and NEMO D311N do not increase linear ubiquitination at Htt-polyQ aggregates. HEK293T cells were transfected with Htt-Q97-GFP and either wildtype HA-NEMO, HA-NEMO Q330X, or HA-NEMO D311N. The cells were lysed 72 h after transfection under denaturing conditions in 1.5 % (w/v) SDS. SDS-insoluble pellets were dissolved in formic acid. Formic acid-dissolved aggregates and the SDS-soluble fractions were analyzed by immunoblotting using antibodies against M1-linked ubiquitin, Htt, NEMO, and β-actin.

Bacteria that escape from the vacuolar compartment into the cytosol, such as *Salmonella* species, are coated by ubiquitin to restrict bacterial proliferation[62–64]. In this pathway, the RNF213 ubiquitin ligase ubiquitylates the bacterial outer membrane component lipopolysaccharide (LPS), which is a prerequisite for the subsequent recruitment of LUBAC[64]. HOIP binds to pre-existing ubiquitin at the bacterial surface and assembles M1-linked ubiquitin chains on bacterial membrane proteins or on ubiquitin moieties previously attached by RNF213. Linear ubiquitin chains then recruit Optineurin and NEMO to the bacterial surface, which both bind to M1-linked ubiquitin with high affinity via their UBAN domain. Whereas Optineurin induces selective autophagy to promote the clearance of bacteria, NEMO locally activates the IKK complex and thereby transforms the bacterial surface into an NF-κB signaling platform[62,63]. LUBAC-mediated aggrephagy differs from bacterial xenophagy in some substantial aspects. First, HOIP is recruited to cytosolic bacteria via its NZF domain by pre-existing ubiquitin assembled by RNF213[64], whereas VCP/p97 is required to recruit HOIP to misfolded proteins by an interaction of the PIM domain of VCP/p97 with the PUB domain of HOIP[49]. Second, Optineurin is required for anti-bacterial autophagy, whereas for the clearance of misfolded aSyn p62 plays a major role, though we cannot exclude an additional role of Optineurin. Third, NEMO is apparently dispensable for bacterial autophagy[62] and mitophagy[74,96], but required for autophagic degradation of aSyn. This difference may be attributable to the fact that p62 is a key cargo receptor in aggrephagy. The increased abundance of p62 in the brain of the Q330X NEMO patient may reflect either a compensatory upregulation by transcription factors other than NF-κB, such as NRF2[97], or an accumulation of p62 due to a decreased autophagic flux.

Even though the role of LUBAC and NEMO in protein quality control seems to be independent of NF-κB signaling, impaired activation of the NF-κB prosurvival pathway may contribute to the toxicity of protein aggregates. Whereas prolonged and excessive NF-κB activation is associated with inflammation, constitutive NF-κB

signaling in neurons regulates synaptic plasticity and neuronal viability[98–102]. Moreover, induced NF-κB activation protects from neuronal cell death in various stress conditions[102–105]. Of note, several neurotrophic proteins, such as NGF, BDNF, and GDNF, signal via NF-κB to promote neuronal viability[106–110]. Our data indicate that similar to intracellular bacteria, an NF-κB signaling platform is assembled at aSyn aggregates. However, this signaling platform is not functional, since p65 seems to be sequestered and trapped at the aggregates so that not even an additional NF-κB-activating stimulus, like TNF, promotes nuclear translocation of p65. Thus, prosurvival signals cannot efficiently be transduced via NF-κB in cells with aSyn or Htt-polyQ aggregates. It is noteworthy in this context that our study revisits an earlier finding linking NEMO to PD. We previously discovered that linear ubiquitination of NEMO is a prerequisite for Parkin to prevent stress-induced neuronal cell death. This activity of Parkin was associated with adding K63-linked ubiquitin to NEMO, suggesting that Parkin can act as a priming E3 ligase for subsequent modification of NEMO by LUBAC[111,112]. Whether Parkin also plays a role in LUBAC-mediated protein quality control is an interesting question to be addressed in further studies.

Finally, our study adds to the notion that different neurodegenerative diseases share common pathways. The accumulation of various proteins linked to neurodegeneration in the brain of the Q330X NEMO patient reflects a general neuronal proteostasis dysregulation, which at least partially can be explained by defective p62-mediated selective autophagy. The presence of α-synuclein, tau, and TDP-43 aggregates has also been reported in some patients with *OPTN* mutations affecting the UBAN domain of Optineurin, which binds linear ubiquitin chains[113]. Thus, neurodegeneration in these patients may also be linked to defective protein quality control downstream of LUBAC. Shared pathological mechanisms related to proteostasis dysregulation may entail specific targets for disease-modifying strategies. Further studies need to address whether the linear ubiquitination machinery can be exploited for therapeutic approaches.

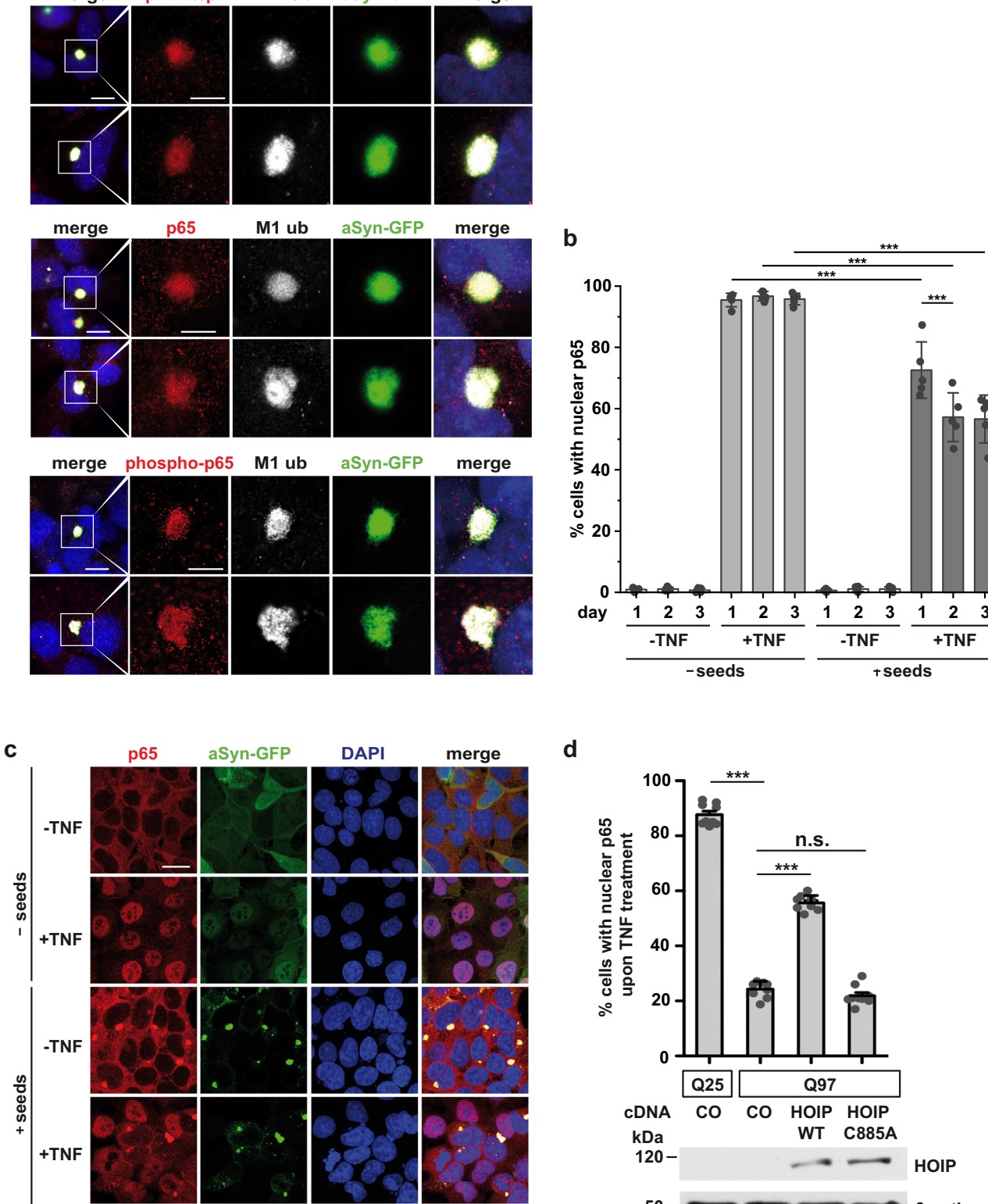

## Methods

### Ethical statement

Research performed for this study complies with all ethical regulations of the respective local authorities the Charité Universitätsmedizin Berlin (Clinical Ethics Committee), and the Institute for Neuropathology, University Medical Center Hamburg-Eppendorf (Clinical Ethics Committee), Hamburg, Germany. *Post mortem* brain tissue from the index case with the NEMO mutation and age-matched controls from the same brain regions were obtained from the Autopsy Service in the Department of Pathology at the University of California San Francisco. Brain tissue samples were collected after informed consent was obtained from the patient or their families, in accordance with guidelines put forth in the Declaration of Helsinki. Autopsy consent and all protocols were approved by the Human Gamete, Embryo, and

**Fig. 5 | A local NF-κB signaling platform is assembled at aSyn aggregates that does not promote nuclear translocation of p65. a** IKKα/β and p65 are recruited to aSyn aggregates. SH-SY5Y cells stably expressing aSyn A53T-GFP were treated with aSyn A53T seeds, fixed on day 3 after seeding, and analyzed by immunocytochemistry and fluorescence SR-SIM using antibodies against phospho-IKKα/β, p65, phospho-p65, and M1-linked ubiquitin. Scale bar, 10 μm (overview) and 5 μm (inset). **b, c** Assembling of an NF-κB signaling platform at aSyn aggregates does not result in p65 nuclear translocation and impairs TNF-induced NF-κB activation. **b** Samples were prepared as described in a, treated with TNF for 15 min (25 ng/ml) on day 1, 2, and 3 after seeding, fixed and quantified for nuclear translocation of p65 by immunocytochemistry and fluorescence microscopy using antibodies against p65. For the seeded samples, only cells with aSyn aggregates were used for quantification. Data represent the mean ± SD of five independent experiments. Statistics:

One-way ANOVA followed by Tukey's multiple comparison test. ***$p \le 0.001$. **c** Representative immunofluorescence images of the experiment described in b (day 2 after seeding). Scale bar, 20 μm. **d** TNF-induced p65 nuclear translocation is impaired in cells with Htt-polyQ aggregates. SH-SY5Y cells were transiently transfected with GFP-tagged Htt-25Q or Htt-Q97 and either vector (co), wildtype HOIP or catalytically inactive C885A HOIP, as indicated. On day 3 after transfection, the cells were treated with TNF (20 ng/ml, 20 min), and nuclear translocation of p65 was analyzed as described in (**b**). Expression of HOIP was analyzed by immunoblotting, actin was used as input control. Data represent the mean ± SD of three independent experiments each performed in triplicates. At least 600 transfected cells were assessed per condition. Statistics: One-way ANOVA with Tukey's Multiple Comparison posthoc test; ***$p \le 0.001$.

Stem Cell Research Committee and the Institutional Review Board (IRB) at UCSF.

## Human brain sections

*Post mortem* brain samples from the Q330X NEMO patient, from DLBD patients and from respective controls were provided by the Department of Pathology, University of California, San Francisco, California, USA. PD brain samples were provided by the Charité Universitätsmedizin Berlin, Germany. *Post mortem* brain samples of frontal isocortex from neuropathologically confirmed AD, DLBD, FTLD and tauopathy patients were obtained from the Institute of Neuropathology, University Medical Center Hamburg-Eppendorf, Hamburg, Germany. Available data are listed in the Supplementary Table 1.

## Genetic analyses

For whole-genome sequencing (WGS) performed at CENTOGENE GmbH, genomic DNA was fragmented by sonication, and Illumina adapters were ligated to the generated fragments for subsequent sequencing on the HiSeqX platform (Illumina Inc., San Diego, CA). The achieved average coverage depth was at least 30x. Data analysis included base calling, de-multiplexing, and alignment to the hg19 human reference genome (Genome Reference Consortium GRCh37). The WGS finding was confirmed by Sanger sequencing of the relevant IKBKG exon on a ABI 3500xL Genetic Analyzer (Applied Biosystems) and analyzed using GeneMapper software (Applied Biosystems). The sequence variant nomenclature followed the standard Human Genome Variation Society guidelines (http://varnomen.hgvs.org/). Variants were classified according to the guidelines of the American College of Medical Genetics (ACMG)[114]. Written informed consent to participate in this study was provided. In addition, we obtained consent from the husband of the deceased patient to publish information that identifies individuals (including three or more indirect identifiers).

## DNA constructs

The following constructs were described previously: human HOIP, HOIL-1L, SHARPIN, HA-HOIP, HOIL-1L-HA, SHARPIN-HA, HA-HHARI, FLAG-NEMO WT, FLAG-NEMO D311N, HA-IKKβ, mCherry, NF-κB-luciferase reporter[111]; HA-HOIP C885A, Htt-Q25-GFP, Htt-Q97-GFP[49]; p62 and p62ΔUBA[12]. HA-NEMO WT and HA-NEMO D311N were generated using the following primers: HA-NEMO-fwd: 5′-ATATGGATCC AATAGGCACCTCTGGAAGAGC-3′, HA-NEMO-rev: 5′-ATATGCGGCCGC CTACTCAATGCACTCCATGACATG-3′. The amplified fragment was digested with BamHI and NotI and cloned into pcDNA3.1-N-HA. HA-NEMO Q330X was generated using the following primers: HA-NEMO Q330X-fwd: 5′- ATAT GGATCC AATAGGCACCTCTGGAAGAGC-3′, HA-NEMO Q330X-rev: 5′-ATATGCGGCCGCTCACAGGAGCTCCTTCTTC TCGG-3′. The amplified fragment was digested with BamHI and NotI and cloned into pcDNA3.1-N-HA. FLAG-NEMO Q330X was generated using the following primers: FLAG-NEMO Q330X-fwd 5′- GCGC AAGCTT ATGGACTACAAGGATGATGATGACAAG-3′, FLAG-NEMO

Q330X-rev 5′- ATAT TCTAGA CTACAGGAGCTCCTTCTTCTCGGC-3′. The amplified fragment was digested with HindIII and XbaI and cloned into pEF4 -N-FLAG. HA-IκBα was generated using the following primers: HA-IκBα-fwd: 5′- ATAT GGATCC ACCGAGGACGGGGACTCG-3′, HA-IκBα-rev: 5′- ATAT GCGGCCGC CTATAACGTCAGACGCTGGCC-3′. The amplified fragment was digested with BamHI and NotI and cloned into pcDNA3.1-N-HA. p62ΔUBA-HA (AA 1-388) was generated using the following primers: p62ΔUBA-HA-fwd: 5′- ATAT GAATTC GCCACC ATG GCG TCG CTC ACC GTG-3′, p62ΔUBA-HA-rev 5′- TATAGCGGCCGC T GGC GGG AGA TGT GGG TAC-3′. The amplified fragment was digested with EcoRI and NotI and cloned into pcDNA3.1-C-HA. aSyn A53T-GFP was generated using the following primers: α-Synuclein A53T-eGFP-fwd: 5′- ATAT AAGCTT GCCACC ATG GAT GTA TTC ATG AAA GGA C-3′, α-Synuclein A53T-eGFP-rev: 5′- ATATGCGGCCGCGGCTTCAGGTTCG-TAGTC-3′. The amplified fragment was digested with HindIII and NotI and cloned into pcDNA3.1-C-eGFP.

## Cell lines

HEK293T cells (CRL-1573; American Type Culture Collection) were cultured in Dulbecco's modified Eagle's medium (DMEM) supplemented with 10% (v/v) fetal bovine serum (FBS) and 100 IU/ml penicillin 100 μg/ml streptomycin sulfate. SH-SY5Y (DSMZ number ACC 209), were cultured in Dulbecco's modified Eagle's medium F-12 (DMEM/F-12) supplemented with 15% (v/v) fetal bovine serum (FBS), 100 IU/ml penicillin 100 μg/ml streptomycin sulfate and 1% nonessential amino acids. Mouse embryonic fibroblasts (MEFs) derived from wildtype or NEMO KO mice[115] or p62 KO mice[116] were cultured in Dulbecco's modified Eagle's medium (DMEM) supplemented with 10% (v/v) fetal bovine serum (FBS) and 100 IU/ml penicillin 100 μg/ml streptomycin sulfate.

## Generation of NEMO CRISPR/Cas9 knockout (KO) SH-SY5Y cells and HOIP CRISPR/Cas9 KO HeLa cells

sgRNAs (RNF31-24147982 AGGGUGUUGAGGUAGUUUCG; RNF31-24147993 GAGCCGUGGACAGGGUGUUG; IKBKG-154552050 UGUGAG AUGGUGCAGCCCAG; IKBKG-154552185 GAGGAGAAUCAAGAGCUC CG) were designed using the Synthego website (www.design.synthego. com). 1.5 nmol sgRNAs were rehydrated in 50 μl nuclease-free 1× TE buffer (10 mM Tris–HCl, 1 mM EDTA, pH 8.0) to a final concentration of 30 μM (30 pmol/μl). sgRNA and recombinant CAS9 were delivered as ribonucleoprotein (RNP) complexes using a 4D-Nucleofector X-Unit (Lonza). Briefly, for the assembly of the RNP complexes, Cas9 2NLS and sgRNAs were combined in Nucleofector™ solution at a molar ratio of 9:1 sgRNA to Cas9 and incubated for 10 min at room temperature. The cells were resuspended at a concentration of 150,000 cells/5 μl. 5 μl of the cell suspension was added to the 25 μl of pre-complexed RNPs for a total transfection volume of 30 μl per reaction and transferred to Nucleofector cartridges. Nucleofection was performed according to the predefined protocol (CA-137 for SH-SY5Y- and CN-114 for HeLa cells) and cells were carefully resuspended in each well of the Nucleocuvette™ with 70 μl of pre-warmed growth medium and

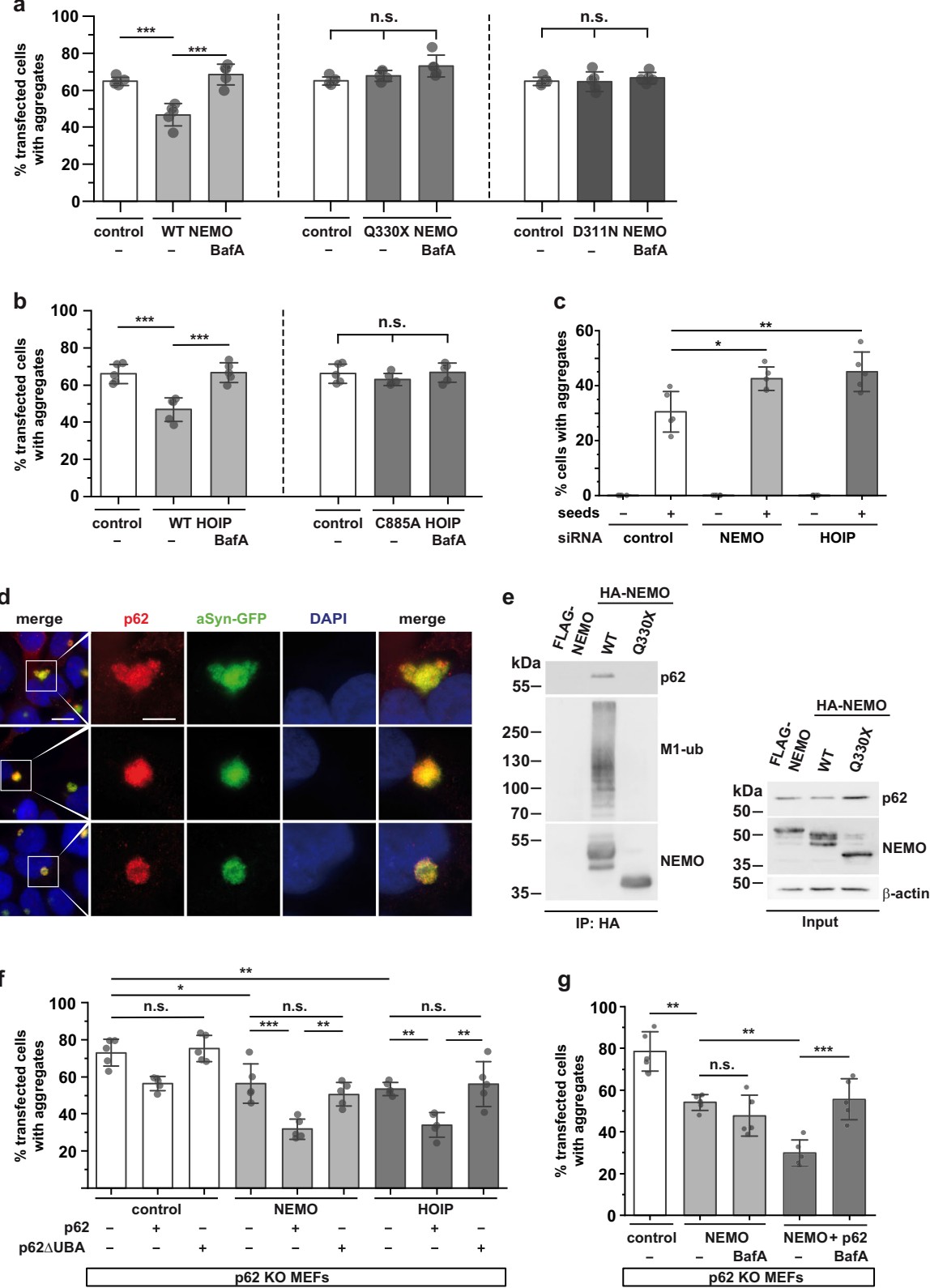

transferred to the pre-warmed 6-well and incubated in a humidified 37 °C/5% $CO_2$ incubator. After 24 h the medium was replaced.

For clone screening, the cells were split into two 6-well cell culture plates, and pools were analyzed by PCR and subsequent DNA sequencing. For this, primer pairs (HOIP_fwd: AGTCCCACCCTCT CTCCTAG, HOIP_rev: TGTGACTGTAGCAACCTGGT, NEMO_fwd: CCT GGAGCTAGGCCTTTTCA, NEMO_rev: ACTTCCTCCCCGCTAATCTG)

were ordered extending approx. 200–250 bp 3′ and 5′ of the sgRNA binding region. To perform cell pool or single clone sequencing analysis, genomic DNA was isolated using a genomic DNA extraction kit (Monarch Genomic DNA Purification Kit, New England Biolabs, Frankfurt am Main, Germany) and the PCR was optimized to yield a single amplicon. Following PCR product purification (NucleoSpin Gel and PCR Clean-up, Macherey-Nagel GmbH, Düren, Germany), the DNA

**Fig. 6 | NEMO decreases the number of cells with aSyn aggregates in a p62-dependent manner. a** WT NEMO but neither Q330X NEMO nor D311N NEMO decreases the number of cells with aSyn aggregates. SH-SY5Y cells stably expressing aSyn A53T-GFP were transiently transfected with either WT HA-NEMO, Q330X HA-NEMO, D311N HA-NEMO, or mCherry as a control and treated with aSyn A53T seeds 24 h after transfection. The cells were treated 16 h after seeding with Bafilomycin A1 (BafA, 25 nM). 40 h after seeding, the cells were fixed and analyzed by immunocytochemistry and fluorescence microscopy using anti-HA antibodies. The fraction of HA- or mCherry-positive cells containing aSyn-GFP aggregates was quantified. Data are shown as mean ± SD based on 5 independent experiments. At least 750 cells per condition were quantified. Statistics were applied to the entire dataset. For better comparability, the control is shown for each NEMO construct. Statistics: One-way ANOVA followed by Tukey's multiple comparison test. ***$p \leq 0.001$. **b** Catalytically active HOIP decreases the number of cells with aSyn aggregates. SH-SY5Y cells stably expressing aSyn A53T-GFP were transiently transfected with either WT HA-HOIP, catalytically inactive C885A HA-HOIP, or mCherry as a control, and treated as described in a. The fraction of HA- or mCherry-positive cells containing aSyn-GFP aggregates was quantified. Data are shown as mean ± SD based on 5 independent experiments. At least 750 cells per condition were quantified. Statistics were applied to the entire dataset. For better comparability, the control is shown for each HOIP construct. Statistics: One-way ANOVA followed by Tukey's multiple comparison test. ***$p \leq 0.001$. **c** NEMO and HOIP silencing increase the number of cells with aSyn-GFP aggregates. SH-SY5Y cells stably expressing aSyn A53T-GFP were transiently transfected with either NEMO- or HOIP-specific siRNAs, treated with aSyn A53T seeds as indicated, and fixed 48 h after seeding. The fraction of cells containing aSyn aggregates was quantified. Data are shown as mean ± SD from 5 biological replicates. At least 750 cells per condition

were quantified. Statistics: One-way ANOVA followed by Bonferroni's multiple comparison test. *$p \leq 0.05$, **$p \leq 0.01$. **d** p62 is present at aSyn aggregates. SH-SY5Y cells stably expressing aSyn A53T-GFP were treated with aSyn A53T seeds, fixed 72 h after seeding, and analyzed by immunocytochemistry and SR-SIM fluorescence microscopy using anti-p62 antibodies. Scale bar, 10 μm (overview) and 5 μm (inset). **e** p62 co-immunoprecipitates with WT NEMO but not with Q330X NEMO. HEK293T cells were transiently transfected with either WT HA-NEMO, Q330X HA-NEMO, or WT FLAG-NEMO to control for unspecific binding. After cell lysis, HA-tagged proteins were immunoprecipitated using anti-HA-beads. Immunoprecipitated proteins were immunoblotted using antibodies against p62, M1-linked ubiquitin, and NEMO. The input was immunoblotted for p62, NEMO, and β-actin. **f** NEMO and HOIP reduce the number of aSyn aggregates in a p62-dependent manner, which requires the UBA domain of p62. p62 KO MEFs were transiently transfected with aSyn A53T-GFP, HA-NEMO, HA-HOIP or mCherry as a control, and p62 or p62ΔUBA, as indicated. One day after transfection, the cells were treated with aSyn A53T seeds, and fixed 48 h after seeding. The fraction of cells containing aSyn aggregates was quantified as described in a. Data are shown as mean ± SD based on 5 individual experiments. Statistics: One-way ANOVA followed by Tukey's multiple comparison test. **$p \leq 0.01$, ***$p \leq 0.001$. **g** The p62-dependent effect of NEMO on aSyn aggregates is sensitive to lysosomal inhibition. p62 KO MEFs were transiently transfected with aSyn A53T-GFP and HA-NEMO, or HA-NEMO and p62, or mCherry as a control, as indicated. The next day, cells were treated with aSyn A53T seeds, and after 16 h treated with Bafilomycin A1 (BafA, 25 nM). 40 h after seeding, cells were fixed and the fraction of cells containing aSyn aggregates was quantified as described in A. Data are shown as mean ± SD based on 5 individual experiments. Statistics: One-way ANOVA followed by Tukey's multiple comparison test. **$p \leq 0.01$, ***$p \leq 0.001$.

was sent for Sanger sequence analysis (Microsynth Seqlab GmbH, Göttingen, Germany). The KO efficiency of the cell pools and single colony clones was determined using the SYNTHEGO ICE analysis website (https://ice.synthego.com). To isolate single KO clones, the KO cell pools were diluted to 1 cell/100 μl and 5 cells/100 μl, and the dilutions were distributed over several 96-well plates. 15–25 clones were grown from single cells and reanalyzed using the above-mentioned process. Finally, clones with a high KO score were amplified and KO efficiency was confirmed by immunoblotting.

### Expression and purification of recombinant proteins

Recombinant α-synuclein expression and seeding was performed as described previously[54,55]. Briefly, α-synuclein (aSyn) A53T encoded on a pT7-7 plasmid was transformed into *E. coli* strain BL21 (DE3). Bacteria were grown in Terrific Broth medium supplemented with ampicillin at 37 °C to a density of an A600 value of 0.8–1.0, and protein expression was induced by adding 1 mM IPTG for 4 h at 37 °C. Bacteria were harvested by centrifugation (6000 rcf, 20 min, 4 °C). Pellet was resuspended in high-salt buffer (750 mM NaCl, 10 mM Tris, pH 7.6, 1 mM EDTA, protease inhibitor tablet (Roche)) and lysed by sonication at 60% power using a probe sonicator (Branson sonifier 450) for a total time of 5 min (30 s pulse on, 30 s pulse off), followed by boiling of the sample for 15 min. After cooling on ice, the sample was centrifuged for 20 min at 6000 rcf. The supernatant was dialyzed overnight with 10 mM Tris, pH 7.6, 50 mM NaCl, 1 mM EDTA. Protein was concentrated with a 3.5 kDa MWCO Amicon ultracentrifuge filter device (Millipore). After filtering the protein through a syringe filter (0.45 μm), the soluble proteins were separated by size exclusion chromatography on a Superdex 200 column (GE Healthcare Life Sciences). 40 2 ml fractions were collected using an ÄKTA chromatography system (GE Healthcare Life Sciences). Collected fractions were analyzed by SDS-PAGE using a 16% (w/v) polyacrylamide-gel followed by Coomassie staining/destaining. Fractions containing aSyn A53T were combined and dialyzed over night with 10 mM Tris, pH 7.6, 25 mM NaCl, 1 mM EDTA. On the next day, the combined fractions were subjected to anion-exchange chromatography on two connected 5 ml HiTrap Capto Q ImpRes (GE Healthcare Life Sciences) anion-exchange columns using a linear gradient, ranging from 25 mM NaCl to 1 M NaCl. Forty 2 ml fractions were collected using an

ÄKTA chromatography system (GE Healthcare Life Sciences). Collected fractions were analyzed by SDS-PAGE using a 16% (w/v) polyacrylamide-gel followed by Coomassie staining/destaining. Fractions containing aSyn A53T were combined and dialyzed over night with 50 mM Tris, pH 7.5, 150 mM KCl. Protein was concentrated with a 3.5 kDa MWCO Amicon ultracentrifuge filter device (Millipore) to a concentration of 5 mg/ml. Aliquots of 1 ml were stored at −80 °C.

To prepare recombinant seeds for the induction of aSyn aggregates in SH-SY5Y cells stably expressing aSyn A53T-GFP, an aliquot containing 1 ml aSyn A53T with a concentration of 5 mg/ml was thawed on ice and centrifuged (20,000 rcf, 30 min, 4 °C). The supernatant was transferred into a new tube and incubated on a thermomixer for 24 h at 37 °C, 900 rpm. The sample was divided into 50 μl aliquots and stored at −80 °C until further use.

pET-Duet1-6xHis-mCherry-p62[71] was recombinantly expressed and purified from *E. coli* Rossetta (DE3) pLysS cells. Bacteria were grown in Luria broth (LB) medium until OD600≈0.6, then induced with 0.3 mM isopropylthiogalactoside (IPTG) and grown at 20 °C for overnight. Harvested cells were resuspended in lysis buffer 50 mM 4-(2-hydroxyethyl)−1-piperazineethanesulfonic acid (HEPES) at pH 7.5, 500 mM NaCl, 10 mM imidazole, 2 mM MgCl$_2$, 2 mM β-mercaptoethanol, complete protease inhibitor, and DNase I and lysed by French press. Lysates were cleared by ultracentrifugation at 40,000 $g$ for 45 min at 4 °C. Supernatants were applied to Nickel-Nitrilotriacetic (Ni-NTA) His-Trap FF column (GE Healthcare) and 6×His-tagged-mCherry-p62 was eluted via a stepwise imidazole gradient (50, 75, 100, 150, 200, and 300 mM). Protein-containing fractions were pooled and dialysed overnight at 4 °C in storage buffer containing 50 mM HEPES pH 7.5, 500 mM NaCl, 2 mM MgCl$_2$, 2 mM β-mercaptoethanol. The protein was filtered using a 0.45 μm syringe, aliquoted, and flash frozen until further use.

Wildtype NEMO-GFP and 4×M1-ubiquitin were expressed and purified as described previously[50]. 8×M1-ub was bought from Enzo Life Sciences.

### Induction of α-Synuclein aggregates by α-Synuclein A53T seeds in cultured cells and primary neurons

SH-SY5Y cells stably expressing aSyn A53T-GFP or MEFs transiently expressing aSyn- A53T-GFP were cultivated on glass coverslips

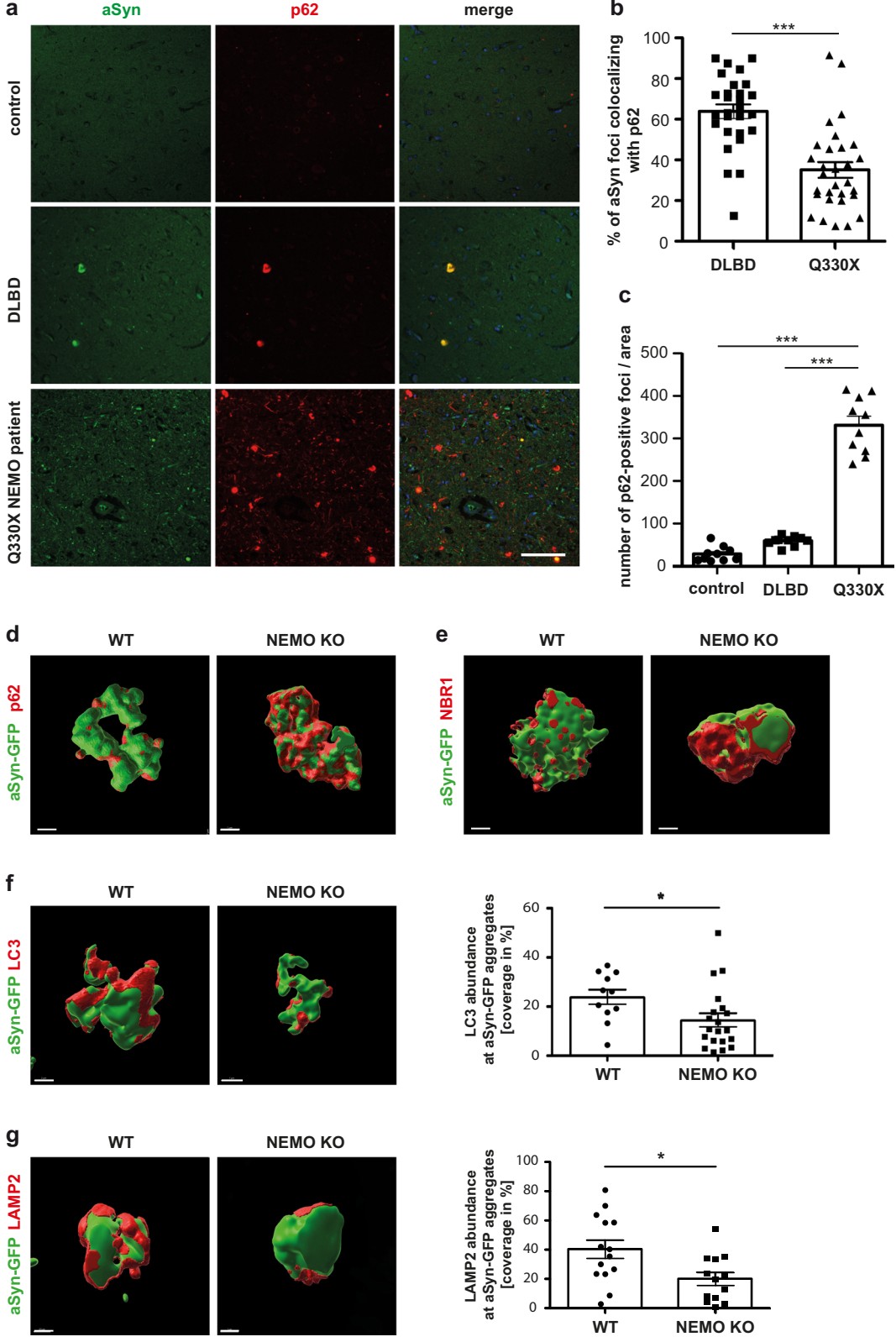

(Laboratory Glassware Marienfeld) for immunofluorescence analysis or on cell culture dishes for biochemical analysis. 24 h after plating, transient transfection, or gene silencing, freshly sonicated aSyn A53T seeds were added to the cells to a final concentration of 12.5 μg/ml as follows:

50 μl of aSyn A53T seeds were thawed at room temperature and added to 950 μl Opti-MEM (Gibco) to obtain a concentration of 250 μg/ml. Sonication was performed with a probe sonicator (Branson sonifier

450) for 3 min (30% power, 10 s intervals). Sonicated seeds were added to 960 μl of Opti-MEM plus 40 μl Lipofectamine 2000 (Invitrogen) in order to increase uptake of seeds by the cells, to a final concentration of 125 μg/ml. After incubation for 15 min at room temperature, the seed solution was added to cells to obtain a final concentration of 12.5 μg/ml per well in Opti-MEM. After 24 h the cells were either harvested, fixed for immunofluorescence experiments, or Opti-MEM was exchanged by Dulbecco's modified Eagle's medium F-12 (DMEM/F-12) supplemented

**Fig. 7 | p62 binds to aSyn aggregates in a NEMO-dependent manner. a, b, c** Colocalization of p62 and aSyn aggregates is decreased in the Q330X NEMO patient brain despite increased p62 expression. **a** Paraffin-embedded brain sections from control, DLBL (Dementia with Lewy Bodies) or the Q330X NEMO patient brain were analyzed by immunohistochemistry and fluorescence SR-SIM using antibodies against aSyn and p62. Scale bar, 200 μm. **b** Foci staining positive for both aSyn and p62 were quantified in 28-30 fields of view per brain section. Data are shown as mean ± SEM, $n = 28/30$ individual cells. **c** Foci staining positive for p62 only were quantified in 10 fields of view per brain section. Data are displayed as mean ± SEM from 10 fields of view. Statistics: Kruskal–Wallis test followed by Tuckey's multiple comparison test. ***$p \leq 0.001$. **d, e** Foci-like concentration of p62 and NBR1 at aSyn aggregates is reduced in NEMO-deficient cells. CRISPR/Cas9 NEMO KO or wildtype (WT) SH-SY5Y cells were transiently transfected with aSyn A53T-GFP. One day after transfection, the cells were treated with aSyn A53T seeds, fixed 48 h after seeding, and analyzed by immunohistochemistry and fluorescence SR-SIM using antibodies against p62 (**d**) and NBR1 (**e**). 3D-reconstructions were performed using the surface module of Imaris 10.0.1 image analysis software. Scale bars, D/E: 1 μm. **f, g** The abundance of LC3 and LAMP2 at aSyn aggregates is decreased in NEMO-deficient cells. CRISPR/Cas9 NEMO KO or WT SH-SY5Y cells were transiently transfected with aSyn A53T-GFP. One day after transfection, the cells were treated with aSyn A53T seeds, fixed 48 h after seeding, and analyzed by immunohistochemistry and fluorescence SR-SIM using antibodies against LC3 (**f**) and LAMP2 (**g**). 3D-reconstructions were performed using the surface module of Imaris 10.0.1 image analysis software. Coverage of aSyn-GFP aggregates by LC3 and LAMP2 was analyzed using the Imaris 10.0.1 surface modules and a surface-to-surface MatLab Plugin. The coverage of the reconstructed aSyn-GFP surface was quantified and plotted as percentage of the total aSyn-GFP aggregate surface for LC3 (**f**) and LAMP2 (**g**). Data are displayed as mean ± SEM, $n = 11/20$ individual cells for LC3 (*$p = 0.0422$) and $n = 14/13$ individual cells for LAMP2 (*$p = 0.0143$). Statistics: two-tailed student's *t*-test. *$p \leq 0.05$. Scale bars, **f** 2 μm (left panel), 1 μm (right panel), **g** 0.5 μm (left panel), 1 μm (right panel).

with 15% (v/v) FCS, 1% (v/v) Penicillin/Streptomycin (Gibco) and 1% (v/v) minimum essential medium non-essential amino acids (Gibco), and cells were further grown for 24 to 48 h prior to harvesting or fixing.

For the induction of aSyn aggregation in primary cortical mouse neurons by aSyn A53T seeds, cells were plated on poly-L-lysine- and laminin-coated coverslips, and sonicated aSyn A53T seeds were added on DIV 5 to a final concentration of 1 μg/ml. Primary neurons were fixed with 4% PFA 7 or 10 days after seeding and prepared for immunocytochemistry.

**Detergent solubility assay**

SH-SY5Y cells stably expressing aSyn A53T-GFP were grown in 3.5 cm dishes and harvested in cold PBS 24, 48, or 72 h after seeding with aSyn A53T seeds. Cells were lysed 10 min on ice in 1% (v/v) Triton X-100 in TBS supplemented with protease inhibitor (cOmplete, Roche) and phosphatase inhibitor (PhosStop, Roche). Samples were centrifuged (15 min, 20,000 rcf, 4 °C), the supernatants were transferred into new tubes, and 5× Laemmli sample buffer was added (1% Triton X-100-soluble fraction). The pellets were washed two times with lysis buffer, solved in 1% (w/v) SDS in TBS, and boiled 15 min at 99 °C. In addition, DNA was sheared by passing the samples 15 times through a 23-Gauge needle. Finally, 5× Laemmli sample buffer was added (1% SDS-soluble fraction). Equal amounts of the two fractions were used for analysis by immunoblotting using the indicated antibodies.

**Immunocytochemistry**

Cells were cultivated on glass coverslips (Laboratory Glassware Marienfeld). For some experiments, coverslips were coated with both poly-L-lysine (PLL) (Sigma) and laminin (Sigma) or PLL only. 24–72 h after seeding with aSyn A53T seeds, the cells were fixed for 10 min with 4% paraformaldehyde in PBS or Tris pH 7.4, and permeabilized and blocked in 0.2% (v/v) Triton X-100, 5% (v/v) goat serum in PBS or Tris for 2 h. Cells were stained with primary antibodies (Table 1) at a dilution of 1:100–1:1000 in 0.2% (v/v) Triton X-100, 5% (v/v) goat serum in PBS or Tris at 4 °C overnight, washed 3× with PBS or Tris, and incubated with fluorescent dye-conjugated secondary antibodies Alexa Fluor 488, 555, or 647 (Thermo Scientific), at a dilution of 1:1000 for 1 h at room temperature. After extensive washing, cells were mounted in Fluoroshield G (Thermo Scientific) with DAPI (Sigma).

**Immunohistochemistry**

For peroxidase immunohistochemistry, paraffin-embedded brain sections (5 μm) were deparaffinized with xylene and ethanol and briefly washed with deionized water. Brain sections were cut from archived paraffin tissue blocks obtained during routine brain autopsy after whole-brain fixation in formalin for at least two weeks. Antigen retrieval through microwaving in 100 mM citrate buffer pH 6.0 was followed by blocking of endogenous peroxidase with 5% $H_2O_2$ in methanol. Then sections were transferred in PBS with 0.02% Brij35 and blocked with 2% FBS in PBS. Incubation with the primary antibody (Table 1) was performed overnight at 4 °C. After rinsing with 0.02 % Brij35 in PBS, antibody binding was detected and enhanced by DCS Super Vision 2 HRP-Polymer-Kit (DCS, Germany) using the chromogen DAB. Counterstaining with hematoxylin for cellular structures was performed. Microscopic images were obtained with a BX50 microscope and Cell-D software (Olympus).

For immunofluorescence histochemistry, paraffin-embedded human brain sections were deparaffinized, and antigen retrieval was performed as described above. Sections were blocked and permeabilized with 10% (v/v) goat serum, 0.3% (v/v) Triton X-100 in PBS for 1 h. Primary antibodies at a dilution of 1:50 (α-synuclein), 1:100 (Tau, TDP-43, M1Ubi, and NEMO) or 1:250 (p62) in 10% (v/v) goat serum, 0.3% (v/v) Triton X-100 in PBS were incubated at 4 °C for 48 h. After the brain sections were washed 3x with PBS and blocked with 2% (w/v) BSA in PBS for 1 h, they were incubated with fluorescent dye-conjugated secondary antibodies Alexa Fluor 488 and 555 (Thermo Scientific) at a dilution of 1:1000 in 2% (w/v) BSA in PBS at room temperature for 1 h. After extensive washing, sections were mounted with Prolong Gold including DAPI (Thermo Fisher Scientific). Confocal images were obtained using a Zeiss ELYRA PS.1 equipped with an LSM880 (Carl Zeiss, Oberkochen). Super-resolution and confocal images were processed using the ZEN2.1 software (Carl Zeiss, Oberkochen).

**Fluorescence microscopy**

Fluorescence microscopy was performed using a Zeiss ELYRA PS.1 system equipped with an LSM880 (Carl Zeiss, Oberkochen) and a 20x/0.8, 63x/1.4 oil or 100x/1.46 oil immersion objective or a C2+ system (Nikon). Super-resolution images were generated by the Structured Illumination (SIM) technology using 405, 488, 561, and 647 nm widefield laser illumination. SIM confocal images were processed using the ZEN Black software (Carl Zeiss, Oberkochen), image data were exported using the ZEN Blue software for further use. For the analysis of stained human brain sections, laser scanning microscopy was performed using the 405, 488, 561, and 647 nm laser illumination set in individual channels to avoid cross-talk. The pinhole was adjusted to generate optical section of 2–5 μm, the acquisition settings were kept constant throughout the experiment. For the analysis of p62 positive aSyn aggregates in human brain sections, a 2 × 2 tile scan using the 20x/0.8 objective acquired and subsequently stitched with ZEN Black software.

**Colocalization studies.** To investigate the colocalization of aSyn, NEMO and M1-ub, the respective cell lines were transiently transfected with pcDNA3.1 aSyn A53T-GFP and seeded with aSyn seeds as described above. After 48 h the cells were fixed and stained with the respective antibodies as described under Immuncytochemistry.

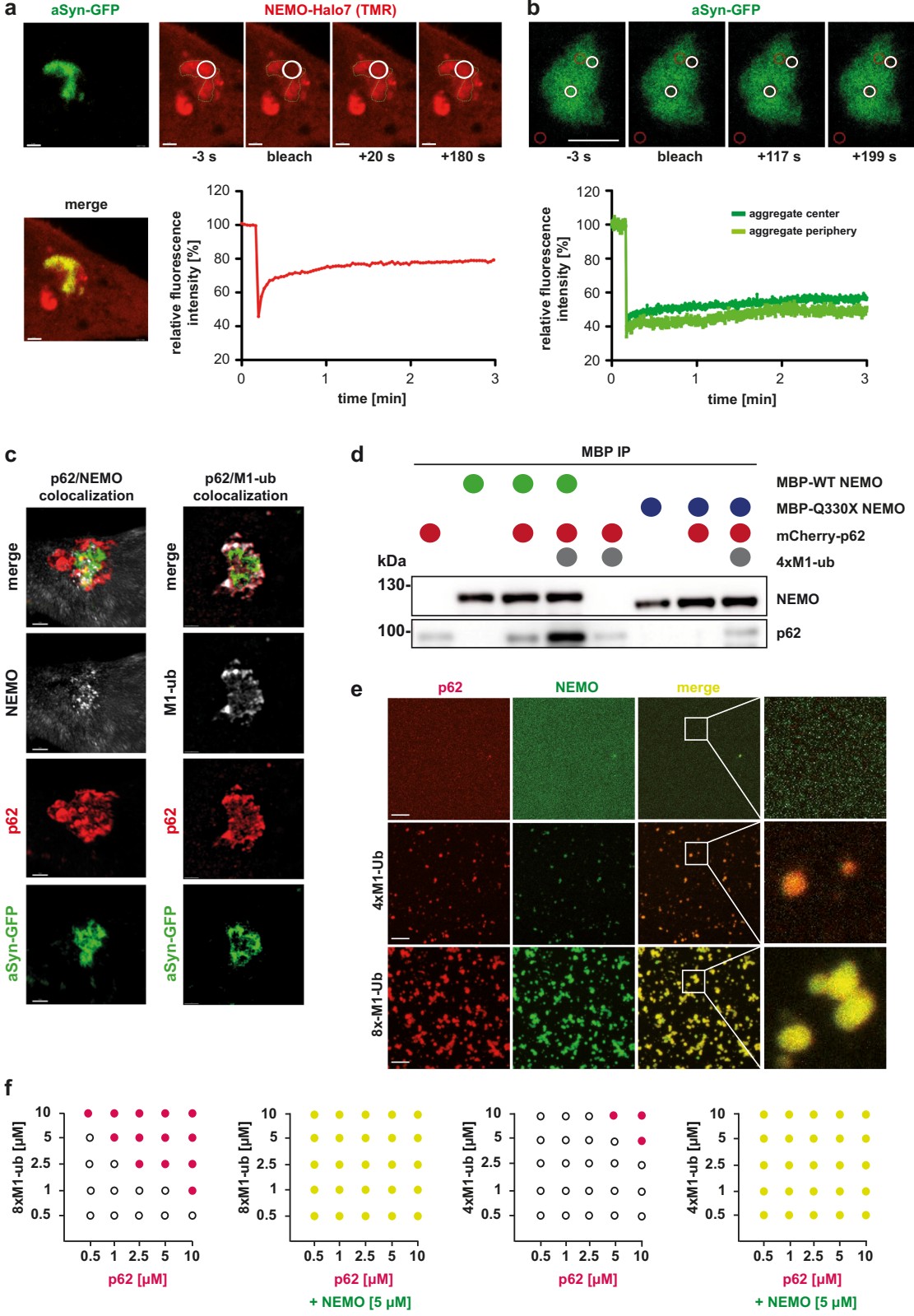

To quantify the colocalization of NEMO and aSyn, M1-ub and NEMO, and M1-ub and aSyn, samples were imaged with constant laser settings, and respective signal thresholds were set to specific signal intensities and kept constant throughout the experiment. A contour was drawn around each protein aggregate and the Pearson colocalization coefficient within the contour was plotted for each comparison.

**Fluorescence recovery after photobleaching (FRAP).** For FRAP analysis, the Halo tag was labeled by incubating the cells for 30 min with 2.5 μM TMR dye and washing for 30 min with cell culture medium prior to live cell imaging. FRAP was performed by 5 consecutive bleaching pulses using the 561 nm laser at 100% intensity within a defined region of interest (white circle) at an aSyn-GFP protein aggregate (green outline). The measurement within the aSyn-GFP

**Fig. 8 | NEMO promotes the local concentration of p62 by co-condensation with M1-linked ubiquitin. a, b** NEMO forms a mobile phase at immobile aSyn aggregates. CRISPR/Cas9 NEMO KO HeLa cells were transiently transfected with aSyn A53T-GFP and NEMO-Halo7. One day after transfection the cells were incubated with aSyn A53T seeds for 48 h. **a** For labeling of the Halo tag, the cells were incubated for 30 min with 2.5 μM TMR dye and washed for 30 min with cell culture medium prior to live cell imaging. Fluorescence recovery after photobleaching (FRAP) was performed by 5 consecutive bleaching pulses using the 561 nm laser at 100% intensity within a defined region of interest (white circles) at an aSyn-GFP aggregate (green). Fluorescence recovery was measured for 3 min and plotted as a percentage of baseline fluorescence. Scale bar, 1 μm **b** aSyn-GFP aggregates were bleached at two independent positions at the aggregate center or aggregate periphery (white circles) using 5 consecutive bleaching pulses using the 488 nm laser at 100% intensity within defined regions of interest. Fluorescence recovery was measured for 3 min and plotted as a percentage of baseline fluorescence. Scale bar, 5 μm. **c** p62 colocalizes with NEMO and M1-linked ubiquitin at aSyn aggregates. HeLa cells were transiently transfected with aSyn A53T-GFP. One day after transfection, the cells were treated with aSyn A53T seeds, fixed 48 h after seeding and analyzed by immunohistochemistry and fluorescence SR-SIM using anti-antibodies against p62, NEMO and M1-ubiquitin (all at endogenous expression). Shown are

representative images of p62 co-localizing with NEMO (left panel) and p62 co-localizing with M1-ubiquitin (right panel) at aSyn-GFP aggregates. Scale bar, 1 μm. **d** The interaction of NEMO with p62 is enhanced in presence of linear ubiquitin chains. 2.5 μM recombinant MBP-NEMO WT or Q330X and 2 μM recombinant mCherry-p62 were incubated with or without 1 μM recombinant linear tetra-ubiquitin (4×M1-ub) with 25 μL MBP affinity agarose beads. Proteins were immunoblotted using antibodies against p62, and NEMO. **e** p62 and NEMO co-condensate in the presence of M1-linked polyubiquitin. 2.5 μM recombinant mCherry-p62 (red) mixed with 5 μM recombinant wildtype NEMO-GFP (green) were supplemented with M1-linked ubiquitin to induce phase separation. Top lane: No M1-linked ubiquitin added. Middle lane: 2.5 μM M1-linked tetra-ubiquitin (4×M1-ub). Bottom lane: 1.0 μM M1-linked octa-ubiquitin (8×M1-ub). Shown are laser scanning microscopy images. Scale bar, 10 μm. **f** NEMO reduces the threshold concentrations required for ubiquitin-dependent p62 phase separation. Phase diagrams depicting concentration-dependent phase separation of p62 and M1-linked octa- or tetra-ubiquitin (8×M1-ub or 4×M1-ub) with or without 5 μM recombinant wildtype NEMO. p62 was incubated in presence of recombinant M1-linked ubiquitin at the concentrations indicated and analyzed by laser scanning microscopy. Black empty circles: no phase separation; red/yellow solid circles: phase separation.

## Table 1 | Antibodies used in this study

| Antibody | Source | Identifier |
| --- | --- | --- |
| Mouse monoclonal anti-HA | BioLegend | Cat# 901502, RRID: AB_2565007 |
| Rabbit polyclonal anti-IKBKG/NEMO | Sigma–Aldrich | Cat# HPA000426, RRID: AB_1851572 |
| Mouse monoclonal anti-IKKγ/NEMO | Cell Signaling Technology | Cat#2695, RIDD: N/A, clone: DA10-12 |
| Rabbit anti-IKKγ/NEMO | Abcam | Ab178972 |
| Rabbit monoclonal anti-linear ubiquitin | Millipore | Cat# MABS199, RRID: AB_2576212 |
| Human monoclonal anti-linear ubiquitin | Genentech | clone: 1F11/3F5/Y102L,[56] |
| Mouse monoclonal anti-GFP | Thermo Fisher Scientific | Cat# 14-6674-82, RRID: AB_2572900 |
| Mouse monoclonal anti-α-synuclein | BD Biosciences | Cat# 610787, RRID: AB_398108 |
| Rabbit monoclonal anti-α-synuclein, (phospho Ser129) | Abcam | Cat# ab51253, RRID: AB_869973 |
| Mouse anti-α-synuclein | Santa Cruz Biotechnology | Sc-12767, RRID:AB_628318 |
| Mouse anti-phospho-tau | Invitrogen | Cat# MN1020 |
| Mouse anti-TDP-43 | Proteintech | Cat# 60019-2-Ig |
| Rabbit anti-TDP-43 | Proteintech | Cat#12892-1-AP |
| Mouse monoclonal anti β-actin | Sigma–Aldrich | Cat# A5316, RRID: AB_476743 |
| Rabbit polyclonal anti-phospho-IKK α/β (Ser176, Ser180) | Thermo Fisher Scientific | Cat# 710676, RRID: AB_2532752 |
| Rabbit monoclonal anti-NF-κB p65 | Cell Signaling Technology | Cat# 8242, RRID: AB_10859369 |
| Rabbit monoclonal anti-phospho-NF-κB p65 (Ser536) | Thermo Fisher Scientific | Cat# MA5-15160, RRID: AB_10983078 |
| Rabbit polyclonal anti-p62 / SQSTM1 | MBL International | Cat# PM045, RRID: AB_1279301 |
| Rabbit polyclonal anti-p62 | Proteintech | Cat# 18420-1-AP, RRID:AB_10694431 |
| Mouse monoclonal anti-beta III Tubulin - Neuronal Marker | Abcam | Cat# ab78078, RRID: AB_2256751 |
| Rabbit polyclonal anti-HOIP | Bethyl | Cat# A303-560A, RRID: AB_10949139 |
| Mouse monoclonal anti-IκBα | Cell Signaling Technology | Cat# 4814, RRID: AB_390781 |
| Mouse monoclonal anti-DYKDDDDK Tag (FLAG) | Cell Signaling Technology | Cat# 8146, RRID: AB_10950495 |
| Rabbit monoclonal anti-cleaved caspase-3 | Cell Signaling Technology | Cat# 9664, RRID: AB_2070042 |
| Rabbit polyclonal anti-PARP | Cell Signaling Technology | Cat# 9542 |
| Rabbit monoclonal anti-K63 Ubiquitin | Millipore | Cat# 05-1308, RRID: AB_1587580 |
| Rabbit monoclonal anti-K48 Ubiquitin | Cell Signaling Technology | Cat# 8081, RRID: AB_10859893 |
| Mouse monoclonal anti-Ubiquitin | Santa Cruz Biotechnology | Cat# sc-8017, RRID:AB_628423 |
| Mouse monoclonal anti-LAMP2 | Santa Cruz Biotechnology | Cat# 18822, RRID: AB_626858 |
| Rabbit monoclonal anti-NBR1 | Cell Signaling Technology | Cat# 9891, RRID: AB_10949888 |
| Rabbit polyclonal anti-LC3 | Proteintech | Cat# 14600-1-AP, RRID:AB_2137737 |
| Mouse monoclonal anti-LC3 | Nanotools | Cat# 0260-100/LC3-2G6, RRID:AB_2943418 |
| Rabbit polyclonal anti-huntingtin Exon1 | Enzo Life Sciences | Cat# BML-PW0595-0025, RRID: AB_2051651 |

aggregates was performed using 5 consecutive bleaching pulses with the 488 nm laser at 100% intensity at the indicated regions of interest (white circle). Fluorescence recovery was measured every 20 ms for 3 min and plotted as a percentage of baseline fluorescence.

**3D-image reconstruction and surface coverage analysis.** To investigate the coverage by LC3 and LAMP2 and the spatial localization of p62 and NBR1 at aSyn protein aggregates, image acquisition was performed using a Zeiss Elyra7 equipped with the SIM² module. Images stacks of 3–8 µm in z were collected using the Leap-function and the minimal interval. Following the SIM conversion by ZEN 3.0 black, images were imported into Imaris 10.0.3 image analysis software. 3D-surfaces of aSyn aggregates and the respective antibody signal were reconstructed using the Surface module in batch mode. Surface-to-surface contacts sites were analyzed using a MatLab Plugin (Imaris website modules). The coverage of the reconstructed aSyn aggregate surface by the respective antibody signal was quantified and plotted as the percentage of the total aSyn-GFP aggregate surface.

**Analysis of condensate formation.** Fluorescent imaging laser scanning microscopy was performed using an LSM880 (Carl Zeiss, Oberkochen) with a 63× oil immersion objective. A 63× NA 1.4 oil immersion objective was used to record a z-stack of $67.5 \times 67.5 \times 10$ and 0.330 µm for each optical section. The argon laser power was set to 0.006% at 488 nm with pixel dwell time of 5.71 µs. During all measurements, laser power, gain, and field of view were kept constant. The Z-stacks were then processed to obtain maximum intensity projections.

### Immunoprecipitations

Cells were lysed in 1% (v/v) Triton X-100 in PBS supplemented with 30 mM NEM (Sigma), protease inhibitor (cOmplete, Roche), and phosphatase inhibitor (PhosStop, Roche). The lysates were cleared by centrifugation ($20,000 \times g$, 4 °C, 15 min). Samples were incubated overnight with anti-HA magnetic beads (Thermo Scientific) or for 5 h with GFP-Trap magnetic agarose (Chromotek) at 4 °C under rotation. Beads were washed five times with 1% (v/v) Triton X-100 in PBS. Immunopurified proteins were eluted by adding 2x Laemmli sample buffer and boiling for 10 min at 95 °C. Samples were analyzed by immunoblotting using the indicated antibodies (Table 1).

### p65 translocation assay

SH-SY5Y cells stably expressing α-Synuclein A53T-GFP were plated on cover slips, and after 24 h they were treated with α-Synuclein A53T seeds ( + seeds) or PBS as a control (−seeds). One, two, and three days post-seeding, cells were treated with 25 ng/ml TNF (PeproTech) for 15 min as indicated and fixed with PFA. Samples were analyzed by immunocytochemistry using an antibody for p65. Nuclear translocation was quantified using the green, red, and DAPI channel of a fluorescence microscope (Nikon Eclipse E400).

### Quantification of aSyn A53T-GFP aggregates

After preparation of the samples on coverslips, they were analyzed using the green, red, and DAPI channel of a fluorescence microscope (Nikon Eclipse E400). Cells containing aggregates or fibrillar structures of aSyn A53T-GFP were counted positive, cells with neither aggregates nor fibrillar structures but with cytosolic and nuclear GFP staining were counted negative. When plasmids were transiently transfected during sample preparation, only cells positive for the transfected construct were considered. For quantification of the percentage of cells with aggregates five independent experiments were performed, and at least 150 cells were analyzed per condition for each replicate.

### Analysis of SDS-insoluble Htt-Q97 proteins

HEK293T cells transfected with the indicated plasmids were grown on 10 cm dishes. Three days after transfection, the cells were harvested and lysed under denaturing conditions in TEX buffer (70 mM Tris–HCL pH 6.8, 1.5% SDS (w/v), 20% glycerol (v/v)), vortexed for 10 s, heated to 99 °C, and DNA was sheared by passing the samples 15 times through a 23-Gauge needle. Subsequently, DTT was added to the samples at a final concentration of 50 mM, the samples were boiled for 10 min at 99 °C, and centrifuged for 60 min (20,000 rcf, room temperature). The SDS-insoluble pellets were dissolved in 70 µl 100% formic acid by incubation of the samples for 40 min at 37 °C while shaking at 1000 rpm. Formic acid was evaporated overnight at 30 °C using a Speedvac system (Eppendorf). The remaining protein pellets were solved in Laemmli sample buffer and boiled for 10 min at 95 °C. SDS-insoluble, formic acid-dissolved aggregates were analyzed by immunoblotting using the M1-ubiquitin-specific antibody 1F11/3F5/Y102L.

### Filter retardation assay

To detect Htt aggregates, transfected cells were lysed in 1% (v/v) Triton X-100, 50 mM MgCl$_2$ and 0.2 mg/ml DNase I in PBS. After centrifugation ($180,000 \times g$ for 30 min at 4 °C) the pellet was resuspended in 2% SDS in 100 mM Tris (pH 7.0). After 1 h incubation at room temperature the homogenates were diluted 1:5 in 100 mM Tris (pH 7.0) and filtered through a cellulose acetate membrane with 0.2 µm pore size (GE) using a Slot Blot Blotting Manifold (Hoeffer).

### Sedimentation assay

An aliquot of both aSyn A53T seeds and aSyn A53T monomers was thawed at room temperature. 10 µl of each sample was centrifuged for 10 min at 20,000 rcf, respectively. The supernatant was transferred into a new tube and 10 µl of 2× Laemmli sample buffer was added (sup). The pellets were resuspended in 200 µl PBS and centrifuged again (20,000 rcf, 10 min). Supernatants were discarded and 10 µl PBS and 10 µl 2× Laemmli sample buffer were added to the pellets (pellet). Sup and pellet fractions of both seeds and monomeric aSyn A53T were analyzed by SDS-PAGE and Coomassie staining/destaining.

### Thioflavin T assay

An aliquot of both aSyn A53T seeds and aSyn A53T monomers was thawed at room temperature. For each sample 4 technical replicates were prepared: For each technical replicate either 5 µl of seeds, monomeric aSyn A53T, or PBS were pipetted into a black 96-well plate (Berthold), respectively. Samples were incubated with 95 µl of a 25 µM Thioflavin T (Sigma–Aldrich) in PBS solution for 45 min at room temperature. Fluorometry was performed using a microplate reader (Cytation 5, BioTek, excitation 442 nm, emission 485 nm).

### Atomic force microscopy (AFM)

AFM measurements were conducted on a Bruker Bioscope RESOLV, using Peak Force Tapping Mode at 2 kHz resonant frequency. For liquid AFM measurements a ScanAsyst Fluid+ probe from Bruker Nano was run at 0.5 nN setpoint and a peak force amplitude of 90 nm. Freshly cleaved MICA from PLANO was used as a substrate for in situ measurements. Beam alignment was done in the buffer solution. 4 µl of the target solution was drop cast on the MICA substrate, shortly after (max. 10 s) the volume was filled with 2 ml buffer solution. For dry AFM measurements a ScanAsyst Air probe from Bruker Nano was run at 1.3 nN setpoint and a peak force amplitude of 150 nm. A silicon wafer from PLANO was used as a substrate for dry measurements. 4 µl of the target solution was spin coated on the Si-substrate ($5.59 \times g$) and air counter flux.

### Dynamic light scattering (DLS)

DLS measurements were performed on a Malvern Instruments Zetasizer Ultra (633 nm laser source), in a 20 µl quartz cuvette (ZEN2112). The displayed data were recorded over three cycles in back-scattering mode. The total acquisition time was 2 s. Attenuation and position where fixed by the device automatically. The refractive index and absorbance of polystyrene was used.

## Analysis of proteotoxic stress

**Immunocytochemistry.** MEFs were cultivated on glass coverslips (Laboratory Glassware Marienfeld). 24 h after seeding, cells were heat stressed at 42 °C for 1 h or treated with 0.5 μM MG-132 for 48 h or 100 nM BafA1 for 48 h. Cells were fixed for 15 min with 4 % paraformaldehyde in PBS pH 7.4 and permeabilized with 0.5% (v/v) Triton X-100, 3 mM EDTA pH 8.0, and 5% goat serum in PBS for 30 min at room temperature and then stained with Proteostat® (Enzo Life Sciences, Inc.) at a dilution of 1:2000 in 1× assay buffer (ENZO) for 20 min. Cells were then mounted in Fluorshield with DAPI (Sigma). Fluorescence microscopy was performed using a Zeiss ELYRA PS.1 equipped with an LSM880 (Carl Zeiss, Oberkochen) with a 63 × oil immersion objective. Super-resolution images were generated by the Structured Illumination (SIM) technology. SIM confocal images were processed using the ZEN2.1 software (Carl Zeiss, Jena).

**Trypan blue dye exclusion.** Cells were seeded in a 6-well dish at a density of ~500,000 cells. After 24 h, cells were either heat stressed at 46 °C for 1 h or treated with MG-132 for 16 h (2 μM), or 24 h (2 μM), or 48 h (0.5 μM) or treated with BafA1 for 16 h (500 nM), or 24 h (500 nM), or 48 h (100 nM) and trypsinized. The cell suspension was centrifuged for 5 min at $100 \times g$ and the cell pellets were resuspended in PBS. One part of cell suspension was mixed with one part of trypan blue dye. Trypan blue dye only permeates damaged cell membranes. Therefore, the unstained (viable) and stained (nonviable) cells were counted by using a hemocytometer. To obtain the total number of viable cells per ml of aliquot, the total number of viable cells were multiplied by 2 (the dilution factor for trypan blue). To obtain the total number of cells per ml of aliquot, the total number of viable and nonviable cells were added and multiplied by 2. The percentage of viable cells were calculated as follows: Cell viability (%) = (total number of viable cells per ml of aliquot/total number of cells per ml) *100.

**Immunoblotting.** Proteins were size-fractionated by SDS-PAGE (16% or 8% polyacrylamide) and transferred to nitrocellulose by electroblotting. The nitrocellulose membranes were blocked with 5% non-fat dry milk in TBST (tris-buffered saline (TBS) containing 0.1% Tween 20) for 30 min at room temperature and subsequently incubated with the primary antibody against Poly (ADP-ribose) polymerase (PARP) or active caspase-3 in TBST for 16 h at 4 °C. After extensive washing with PBST, the membranes were incubated with horseradish peroxidase-conjugated secondary antibody for 1 h at room temperature. Following washing with PBST, the antigen was detected with the enhanced chemiluminescence detection system.

## Pull-down assay for protein-protein interaction

2.5 μM recombinant NEMO and 2 μM recombinant p62 were incubated with or without 1 μM recombinant linear tetra-ubiquitin chains with 25 μL MBP affinity agarose beads (Chromotek) for 4 h at 4 °C on rotation. The beads were pre-equilibrated in native buffer containing 50 mM HEPES pH 7.4, 150 mM NaCl, 1 mM EDTA, 1 mM EGTA, 1% Triton X-100, 10% glycerol, 25 mM NaF, 10 μM $ZnCl_2$ supplemented with protease inhibitor (cOmplete, Roche) and N-ethylmaleimid (Sigma) in PBS. After 4 h, the beads were washed 5 times with the same native buffer and then boiled in 2× Laemmli sample buffer for 8 min at 95 °C. Samples were analyzed by immunoblotting using the indicated antibodies (Table 1).

## Phase separation assay

Protein aliquots were thawed on ice. Using Vivaspin 500 columns with 30 or 10 kDa molecular weight cut off (Sartorius Stedim Biotech), the buffer was exchanged to 10 mM Tris, pH 7.4, 1 mM DTT by centrifuging five to eight times for 9 min at 12,000 $g$, 4 °C. After buffer exchange, protein was collected and finally centrifuged at 20,000 $g$ for 10 min at

4 °C to remove aggregates. The final protein concentration was determined by NanoDrop 2000. TEV protease was added to the samples and incubated 1 h for complete cleavage of MBP and 6xHis before microscopy. After the reaction, 10 μl of reaction mix was spotted on ibidi coverslip bottom dishes. The samples were then imaged as maximum intensity projection of a z-stack obtained using laser scanning microscopy.

## Quantification and statistical analysis

Data represent the mean ± SD or SEM, n numbers are indicated in the figure legends. For the quantification analysis in which manual counting was used, not all experiments were performed in a blinded manner. All statistical analyses were performed by using GraphPad PRISM (Version 5; San Diego, CA, USA). To check for Gaussian distribution of the data, the Kolmogorov–Smirnov test was applied. Based on the outcome of the test, appropriate parametric and non-parametric tests were chosen. For the comparison of two independent parametric datasets, the student's t-test was used. For the comparison of more than 2 parametric datasets, one-way ANOVA was applied. To correct for α-error inflation resulting from multiple comparisons, ANOVA was followed by the Tukey's post hoc multiple comparison tests. For the direct comparison of two non-parametric datasets, the Wilcoxon Mann–Whitney (U-test), and for the comparison of more than 2 non-parametric datasets, the Kruskal–Wallis test was used. Significance levels for all tests: $*P \leq 0.05$; $**P \leq 0.01$; $***P \leq 0.001$. For the analysis of microscopic images, representative images are displayed; experiments were performed at least three times with similar results.

## Reporting summary

Further information on research design is available in the Nature Portfolio Reporting Summary linked to this article.

## Data availability

The authors declare that the data supporting the findings of this study are available within the article and its supplementary information files. Source data are provided with this paper. The whole-genome sequencing data generated in this study have been deposited in the NCBI Sequence Read Archive database under accession codes http://www.ncbi.nlm.nih.gov/bioproject/1043442 and https://www.ncbi.nlm.nih.gov/biosample/38340847. Source data are provided with this paper.

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

## Acknowledgements

We thank Tetsuro Ishii for p62 KO MEFs, Marc Schmidt-Supprian for NEMO KO MEFs, Terje Johansen for p62 plasmids, Sascha Martens for mCherry-p62, Yen-Ting Chen for discussing protein adsorption processes, and Genentech for the 1F11/3F5/Y102L antibody. We also thank Bruce Conklin (Gladstone Institutes) and the Gladstone Stem Cell Core for assistance with the patient skin biopsy and fibroblast culture. This work was supported by the German Research Foundation (WI/2111-4, WI/2111-6, WI/2111-8, FOR 2848 to KFW) and Germany's Excellence Strategy EXC 2033 390677874 – RESOLV (to KFW and JT) and EXC 2145 390857198 – SyNergy (to CB and FUH), the NIH (R01 AG065428 to KN) as well as by the joint efforts of The Michael J. Fox Foundation for Parkinson's Research (MJFF) and the Aligning Science Across Parkinson's (ASAP) initiative. MJFF administers the grant ASAP-000282 on behalf of ASAP and itself. SR-SIM microscopy was funded by the German Research Foundation and the State Government of North Rhine-Westphalia (INST 213/840-1 FUGG).

## Author contributions

Conceptualization: N.F., L.A., V.B., S.G., A.S.-V., C.Sachse., C.B., K.N., C.W.C., J.T. and K.F.W. Validation: N.F., L.A., V.B., A.B., S.G., A.S.-V., L.J.K., S.A.C., P.G., V.A.T., E.M.v.W., M.J., A.W., L.J., J.T. and K.F.W. Visualization: N.F., L.A., V.B., A.B., S.G., A.S.-V., V.A.T., M.J., E.J.H., J.T. and K.F.W. Writing—original draft: N.F., V.B., A.B., S.G., A.S.-V., J.T. and K.F.W. Writing – review & editing: L.J.K., S.A.C., K.T., R.B.D., A.W., C.K., C.B., M.G., F.U.H., K.N., C.W.C. and E.J.H. Formal analysis: N.F., L.A., V.B., A.B., S.G., A.S.-V., L.J.K., S.A.C., P.G., E.M.v.W., M.J., A.W. and L.J. Methodology: N.F., L.A., V.B., A.B., S.G., A.S.-V., V.A.T., M.J., A.W., C.K., L.J. and E.J.H. Investigation: N.F., L.A., V.B., A.B., S.G., A.S.-V., L.J.K., S.A.C., P.G., V.A.T., E.M.v.W., M.J., A.W., C.K., L.J. and E.J.H. Data curation: N.F., L.A., V.B. and S.G. Resources: L.J.K., S.A.C., K.T., R.B.D., C.Sachse, G.E., A.K., B.E., C.Saft, C.B., M.G., F.U.H., K.N., C.W.C., E.J.H., R.G. Supervision: J.T. and K.F.W. Funding acquisition: K.F.W. Project administration: J.T. and K.F.W.

## Funding

## Competing interests

R.B.D. is a scientific advisor for Immagene B.V. C.K. serves as a medical advisor to Centogene for the validation of genetic testing reports in the field of movement disorders and dementia, excluding Parkinson's disease, and Retromer Therapeutics and received speakers' honoraria from Bial and Desitin. The remaining authors declare no competing interests.

## Additional information

[1]Department Molecular Cell Biology, Institute of Biochemistry and Pathobiochemistry, Ruhr University Bochum, 44801 Bochum, Germany. [2]Department Biochemistry of Neurodegenerative Diseases, Institute of Biochemistry and Pathobiochemistry, Ruhr University Bochum, 44801 Bochum, Germany. [3]Department of Neurology, St Josef Hospital, Ruhr University Bochum, 44791 Bochum, Germany. [4]Cluster of Excellence RESOLV, 44801 Bochum, Germany. [5]Department of Cellular Biochemistry, Max Planck Institute of Biochemistry, 82152 Martinsried, Germany. [6]Analytical Chemistry II, Faculty of Chemistry and Biochemistry, Ruhr University Bochum, 44801 Bochum, Germany. [7]Department of Biotechnology and Biomedicine, Technical University of Denmark, 2800 Kongens Lyngby, Denmark. [8]Charité - Universitätsmedizin Berlin, Corporate Member of Freie Universität Berlin and Humboldt-Universität zu Berlin, Department of Neuropathology, Charitéplatz 1, 10117 Berlin, Germany. [9]Institute of Neurogenetics, University of Lübeck, Lübeck, Germany. [10]Ernst-Ruska Centre for Microscopy and Spectroscopy with Electrons (ER-C-3/Structural Biology), Forschungszentrum Jülich, Jülich, Germany. [11]Institute for Biological Information Processing (IBI-6/Cellular Structural Biology), Forschungszentrum Jülich, Jülich, Germany. [12]Department of Biology, Heinrich Heine University, Düsseldorf, Germany. [13]Munich Cluster for Systems Neurology, Faculty of Medicine, Ludwig-Maximilians-Universität München, Munich, Germany. [14]Institute of Neuropathology, University Medical Center Hamburg-Eppendorf, Martinistraße 52, 20251 Hamburg, Germany. [15]Munich Cluster for Systems Neurology (SyNergy), 81377 Munich, Germany. [16]Gladstone Institute of Neurological Disease, Gladstone Institutes, San Francisco, CA, USA. [17]Department of Neurology, University of California, San Francisco, CA, USA. [18]Weill Institute for Neurosciences, University of California San Francisco, San Francisco, CA, USA. [19]Department of Pathology, University of California, San Francisco, CA, USA. [20]Present address: Department of Neurology, Klinikum Dortmund, University Witten/Herdecke, 44135 Dortmund, Germany. [21]Present address: Center for Neuropathology and Prion Research, Ludwig-Maximilians University, 81377 Munich, Germany. ✉e-mail: konstanze.winklhofer@rub.de

