## [Peer Review File · Nature Communications]

NEMO reshapes the α -Synuclein aggregate interface and acts as an autophagy adaptor by co-condensation with p62Reviewer #1 (Remarks to the Author):

NEMO reshapes the protein aggregate interface and promotes aggrephagy by co-condensation with p62

In this study, Furthmann et al have discovered an additional function of NEMO that is independent of NF- κ B, namely, its involvement in regulating proteostasis by facilitating the removal of protein aggregates through autophagy. Cells lacking NEMO experience an accumulation of misfolded proteins under conditions of proteotoxic stress, rendering them susceptible to challenges in maintaining proteostasis. Furthermore, they identified a patient carrying a mutation in the NEMO gene (NEMO Q330X), resulting in impaired binding of NEMO to linear ubiquitin chains. This patient displayed a wide-ranging brain proteinopathy characterized by the presence of α -synuclein, tau, and TDP-43 pathology. NEMO enhances the process of linear ubiquitylation in α -synuclein aggregates and facilitates the concentration of p62 in these aggregates. Moreover, NEMO reduces the concentration threshold required for the ubiquitin-dependent phase transition of p62. Concluding, Furthmann et al showed that NEMO plays a crucial role in reshaping the surface of protein aggregates to promote efficient clearance through autophagy. It achieves this by creating a mobile environment that encourages the co-condensation of p62 at the interface of the aggregates.

The data by Furthmann et al make an important contribution to the understanding of p62 mediated phase separation, aggrephagy and NF- κ B-independent roles of NEMO. Additionally, the manuscript is well written, concise, data are represented clearly and conclusions are not overstated. The manuscript can be further improved after addressing the below mentioned suggestions:

Major comments:

1. It is not clear if this mutation of NEMO Q330X occurs in more IP patients or if other patients have similar truncated versions of NEMO (missing the LZ and ZF domains).
2. What are the roles of the LZ and ZF domains in this phenotype? This would be important to know, in order to draw conclusions on other/similar NEMO patient mutations that occur in the C-terminal region. Q330X does not bind M1-UB, although the UBAN domain is intact; nor is it ubiquitylated by HOIP, although the CC2 domain is neither affected. Thus, it would be good to test if the LZ or ZF domains are required for clearing ub-aggregates or for binding p62, to give a hint if other NEMO mutations occurring more C-terminally are also affected.
3. Figure 4A needs a quantification, not just images of 3 cells. Especially since in the 3rd image many cells do not show recruitment of NEMO to α -Syn-GFP.
4. The authors use BafA1 treatment to block p62-mediated aggregate clearance through autophagosome-lysosome degradation, however further experiments are needed to fully prove this. For example: co-staining of aggregates with LC3B or LAMP1 as autophagosome-lysosome markers; in ATG5 or FIP200 KO cells.
5. MG-132 treatments to show increased aggregates in NEMO KO cells should be compared to BafA1 treatment side-by-side.

Minor comments:

Introduction:

6. The role of p62 as an autophagy receptor for aggregates is well established and should already be referred to in the introduction. It might be beneficial for the reader to briefly explain the molecular mechanism of autophagy, in order to understand how p62 removes the protein aggregates.
7. There should be more info on the fact that NEMO, ABINs and OPTN are highly similar in two aspects: 1) they harbor a UBAN domain and have all been identified as autophagy receptors.
8. It should be explained what the LZ and ZF domains in NEMO are and what their functions include, since these domains are missing in the Q330X mutant and are potentially critical for the here discovered role of NEMO.

Figure legends:

9. Figure legends for supplementary figures occur in the main text as well as in the supplementary figure file. One set should be removed.

Figures:

- 13 What is the cell viability in NEMO KO cells upon BafA1 treatment?

Reviewer #2 (Remarks to the Author):

In 'NEMO reshapes the protein aggregate interface and promotes aggrephagy by co-condensation with p62', Furthmann et al. examine how NEMO promotes clearance of misfolded protein aggregates, e.g., α -synuclein aggregates in various cell culture model. It's interesting to note that they claim NEMO facilitates α -synuclein aggregate clearance by encouraging co-condensation with p62. This intriguing study sheds new, perhaps significant light on the process underlying aggrephagy. The authors' intriguing and original work is noteworthy but before I can suggest publication, there are some issues that need to be addressed.

Major issues:

1. The authors claimed that misfolded α -synuclein is decorated with M1-linked ubiquitin and NEMO. In this manuscript, there is minimal proof that α -synuclein is M1 ubiquitinated or that it interacts with NEMO. Comparing the co-localization of NEMO and α -synuclein to that of LUBAC and α -synuclein, the immunofluorescence images in Fig. 3D are ambiguous. The author should evaluate the interaction between α -synuclein aggregates and NEMO.
2. The author should address NEMO-LUBAC dependent ubiquitination of α -synuclein. The IPs are far more convincing evidence.
3. The authors suggested that NEMO causes the co-condensation of α -synuclein aggregates with p62, creating a mobile phase transition that is necessary for effective autophagosomal clearance. It is essential to address the recruitment to LC3 and autophagosome components.
4. Rudenko et al. demonstrated that Δ CT-NEMO stimulates NF- κ B signaling and inflammation. The authors need to discuss about how this manuscript differs from their previous work.

Minor issues:

1. In Figures 1C and D, the authors omitted the control brain.
2. In contrast to other samples, the NEMO fluorescence signal in the PD sample in Figure 1D appears to be stronger. Is the NEMO expression higher in the PD sample?
3. Fig. 1D's depiction of NEMO and α -synuclein colocalization is unclear. In comparison with Tau and TDP-43, NEMO showed only limited colocalization with α -syn in PD.
4. Images of NEMO KO MEF in Fig. 2A cannot show the differences in the percentage of cells with aggregates.
5. Fig. 2C and D are missing every single data point.
6. In Fig. 2F, TNF caused the activation of NF- κ B whereas heat shock to WT HEK293 cells did not. Does NEMO phase separation or ubiquitination alter under these conditions?
7. As previously indicated, the colocalization of α -synuclein and NEMO is still unclear in Fig. 3D. The signals of NEMO were blurry.
8. Colocalization analysis is necessary in Fig. 4A.
9. No fluorescence images for Fig. 4F and G. For evidence of M1 ubiquitination of α -synuclein aggregates in Fig. 4, IP experiments are required.
10. In Fig. 5A, P65 was enriched at α -synuclein aggregates following α -synuclein seeding, whereas in Fig. 2F, P65 was dispersed following heat shock. The potential differences between these two scenarios should be discussed by the authors.

Reviewer #3 (Remarks to the Author):

NEMO is conventionally known and described as a scaffold protein in the NF- κ B transcription factor pathway, where NEMO assembles the kinases (and other proteins) necessary for activation of the pathway. In addition, there are human immunodeficiency diseases that are caused by mutations in the X-linked NEMO gene, and these mutations have generally been described as reducing NEMO's function in NF- κ B signaling. As such, many of NEMO's functions have been described in immune cells

In the paper by Furthmann et al they have identified what appears to be a new NEMO pathway/function, which is independent of NF- κ B, wherein NEMO functions in a proteostasis pathway. In this pathway, NEMO is important for protein aggregate/misfolded protein clearance. Moreover, at least one patient with a NEMO mutation has a widespread brain protein aggregates that involves defective p62 and phase separation. That is, the authors suggest that an additional normal function of NEMO is to promote autophagosomal clearance of protein aggregates by promoting condensation of the liquid-liquid phase separation p62 protein condensates. Thus, these results indicate that NEMO may have a function in neuronal cells and that in certain cases human mutations can also lead to neuronal disease in humans. Overall, this is an extensive series of results with somewhat interesting implications at least in the patient under study.

Concerns

---How good is the evidence that the only mutation in these people that lead to the neuropathy is a NEMO mutation?

---It is unclear to me at what level NEMO protein is expressed in neurons. Is that shown to some extent in Fig. 1C/D? Could the authors investigate the NEMO protein/mRNA expression datasets of NEMO across tissues, esp as in brain compared to immune cells. In particular what does the more neuronal optineurin NEMO-like protein look like in this patient? I believe that OPTN mutations are more common in neuronal protein accumulation diseases than NEMO mutations, and given that some functions/domains of NEMO and OPTN overlap, there is some possibility that some of the findings in this paper are due to OPTN deficiencies that can be, at least partially, substituted by overexpression of NEMO.

--Fig. 1C and D. A few concerns

---I am guessing that the blue staining is DAPI or Hoechst but this is never really described, and is present in some panels but not others (?)

---The co-localization is a bit tenuous. First, wouldn't any cytoplasmic protein "co-localize" with the cytoplasmic NEMO staining? Second, they show single cells in each case; were these selected in some way?

---Fig. 2B, the differences in MG132 treated bars, based on error bars, do not seem especially significant (due to overlap of error bars).

---Fig. 2B, the first two bars are labelled as ns, but it appears that the untreated samples are about 5-10 fold different, which is greater than any of the other pairs. Should the authors simply normalize each same to the untreated control as 1.0, and how would the data look?

---Fig. 2D, the way the error lines above bar graphs are designated do not in any way make it clear what is ns and what is significant. The authors should probably have bracket types of bars to show ns and/or significance. For example, when one looks at the ns bar that is above the first three bars in the graph, one is inclined to believe that the first three bars are not significantly different, but I think they mean that bars 1 and 3 are ns.

----Fig. 3C, the authors do not really describe which scale bars are 5 μ m and which are 20 μ m.

---Fig. 3D, again the overlap with NEMO is not especially convincing, e.g., that it is actually more than what one would see for any cytoplasmic protein

----Fig 4A is potentially an important expt if it shows that WT but not mutant NEMO goes to aggregates. I am guessing the transfection efficiency is not 100% in these experiments. So how does one know that the cells expressing the Syn aggregates are indeed overexpressing the NEMO mutant. I am guessing that is from the first merge low power panels? What do WT NEMO vs mutant NEMO look like in a cell that does not have a Syn aggregate?

---In Fig. 4B, the co-localization of NEMO with the aggregates seems to be around the aggregates. Is that what I am supposed to see?

---The authors show that p62 coats Syn aggregates more robustly in NEMO KO as compared to NEMO WT cells (Fig. 8). Is that what one would expect based on their results? Is it consistent with this sentence: "Our study identified NEMO as a major player in priming the aggregate interphase for p62 condensation."

Rebuttal Letter

We would like to thank the referees for their helpful suggestions and constructive criticism. In the following, we addressed their concerns point by point.

The revised version includes substantial new data and information, based on the reviewers' comments:

- 1.** To address the question of Reviewer #3 whether the Q330X NEMO variant is the only mutation in our patient, we teamed up with geneticists and performed Whole-Genome Sequencing (WGS), using primary fibroblasts from the Q330X NEMO patient. No pathogenic or likely pathogenic variants in genes associated with neurodegenerative disorders were identified (Suppl. Fig. S1). The Q330X mutation occurred sporadically in the patient, confirmed by the mosaicism of wildtype and mutant *IKBKG* gene detected by WGS. Moreover, there was no history of neurodegenerative diseases in the family of the Q330X NEMO patient, supporting a causative role of the NEMO-encoding *IKBKG* gene variant.
- 2.** We used Bafilomycin A1 as an additional stressor in NEMO-deficient cells. In comparison to wildtype cells, NEMO KO cells showed a significant increase in protein aggregates and a decrease in cell viability upon Bafilomycin A1 treatment (Fig. 2B and Fig. S3B).
- 3.** We quantified the colocalization of endogenous NEMO with α -Synuclein (aSyn) aggregates (Fig. 3D) and the colocalization of wildtype and Q330X NEMO with aSyn aggregates (Fig. 4A).
- 4.** As additional evidence for the colocalization of wildtype NEMO with pathogenic protein aggregates, we included a filter retardation assay using cells expressing Huntingtin with an elongated poly-glutamine stretch (Htt-Q97). Whereas wildtype NEMO was retained at the cellulose acetate filter together with Htt-polyQ, Q330X NEMO showed a strongly reduced signal (Fig. 4C).
- 5.** As additional evidence for the effect of wildtype NEMO in amplifying M1-ubiquitylation at protein aggregates, we performed a ubiquitylation assay of cells expressing Htt-Q97. Whereas wildtype NEMO increased the M1-ubiquitin-specific signal in the SDS-insoluble, formic acid-soluble fraction containing aggregates Htt-Q97, Q330X NEMO had no effect (Fig. 4I).
- 6.** We included SH-SY5Y cells silenced for the expression of NEMO or HOIP by RNA interference to corroborate our data from NEMO- or HOIP-overexpressing cells (Fig. 6A, B). Both NEMO-knockdown and HOIP-knockdown cells showed a similar increase in the number of cells with aSyn aggregates in comparison to control cells (Fig. 6C).

7. In addition to p62, we analyzed the localization of NBR1 at aSyn aggregates in wildtype and NEMO KO cells. It turned out that similarly to p62, also NBR1 shows a diffuse and less foci-like pattern of localization at aSyn aggregates (Fig. 7E).

8. The different pattern of p62 and NBR1 localization at aSyn aggregate correlates with a decreased abundance of both LC3 and LAMP2 at the aggregates (Fig. 7F, G).

9. We performed pull-down assays with recombinant p62 and wildtype or Q330X NEMO in the presence or absence of recombinant M1-linked tetra-ubiquitin (4xM1-ub). The interaction between wildtype NEMO and p62 was strongly increased in the presence of 4xM1-ub, whereas the signal for Q330X NEMO, p62 and 4xM1-ub was not increased over background (Fig. 8D).

Reviewer #1:

NEMO reshapes the protein aggregate interface and promotes aggrephagy by co-condensation with p62.

In this study, Furthmann et al have discovered an additional function of NEMO that is independent of NF- κ B, namely, its involvement in regulating proteostasis by facilitating the removal of protein aggregates through autophagy. Cells lacking NEMO experience an accumulation of misfolded proteins under conditions of proteotoxic stress, rendering them susceptible to challenges in maintaining proteostasis. Furthermore, they identified a patient carrying a mutation in the NEMO gene (NEMO Q330X), resulting in impaired binding of NEMO to linear ubiquitin chains. This patient displayed a wide-ranging brain proteinopathy characterized by the presence of α -synuclein, tau, and TDP-43 pathology. NEMO enhances the process of linear ubiquitylation in α -synuclein aggregates and facilitates the concentration of p62 in these aggregates. Moreover, NEMO reduces the concentration threshold required for the ubiquitin-dependent phase transition of p62. Concluding, Furthmann et al showed that NEMO plays a crucial role in reshaping the surface of protein aggregates to promote efficient clearance through autophagy. It achieves this by creating a mobile environment that encourages the co-condensation of p62 at the interface of the aggregates.

The data by Furthmann et al make an important contribution to the understanding of p62 mediated phase separation, aggrephagy and NF- κ B-independent roles of NEMO.

Additionally, the manuscript is well written, concise, data are represented clearly and conclusions are not over-stated. The manuscript can be further improved after addressing the below mentioned suggestions:

Major comments:

1. It is not clear if this mutation of NEMO Q330X occurs in more IP patients or if other patients have similar truncated versions of NEMO (missing the LZ and ZF domains).

The Q330X NEMO mutant has not been identified so far, but other NEMO deletion mutants have been described.

The most common mutation in Incontinentia pigmenti (IP) is a 11.7-kb deletion spanning exons 4 to 10, which is found in approximately 80% of cases¹. This mutation causes embryonic death in males and explains the extremely skewed X inactivation seen in females with IP. Small insertions and deletions, frameshift, missense and nonsense mutations have also been identified^{2,3}. There are several nonsense mutations resulting in the deletion or disruption of the LZ and/or ZF domains, such as Q332X and Q348X^{4,5}. In addition, there are nonsense mutations also affecting the CC2 and/or UBAN domains, such as Q236X, Q239, Y241X, R256X, Q290X, and Y308X^{5,6,7,8}. These mutations have been reported in pediatric patients who are currently probably too young to develop neurodegenerative symptoms. Moreover, the clinical spectrum depends on the extent of functional and genetic mosaicism, resulting from X chromosome inactivation and a co-existence of cells with and without the mutation, since up to 75% of cases occur sporadically. These complex issues explain why patients with similar mutations do not necessarily develop similar phenotypes. However, we think it is important to follow up these NEMO-mutant patients having in mind that they may have an increased risk to develop neurodegeneration.

2. What are the roles of the LZ and ZF domains in this phenotype? This would be important to know, in order to draw conclusions on other/similar NEMO patient mutations that occur in the C-terminal region. Q330X does not bind M1-UB, although the UBAN domain is intact; nor is it ubiquitylated by HOIP, although the CC2 domain is neither affected. Thus, it would be good to test if the LZ or ZF domains are required for clearing ub-aggregates or for binding p62, to give a hint if other NEMO mutations occurring more C-terminally are also affected.

We performed pull-down experiments with recombinant mCherry-p62 and either wildtype MBP-NEMO or Q330X MBP-NEMO in the absence or presence of M1-linked tetra-ubiquitin. The interaction between p62 and NEMO was strongly increased by the presence of both the LZ/ZF domains and M1-linked tetra-ubiquitin (Fig. 8D), suggesting that ubiquitin binding promotes the interaction of p62 and NEMO. In the Q330X NEMO mutant, the impaired interaction between NEMO and p62 is rather caused by the inaccessibility of the UBAN domain due to conformational alterations or defective dimer formation, which turned out to be quite challenging to address for full-length NEMO and Q330X NEMO. We plan to test the individual contribution of the LZ and ZF domain to p62 binding by cloning deletion mutants for recombinant expression. However, as outlined below, the region between the CC2 and LZ form a functional core and any genetic manipulation we performed so far affected binding to M1-linked ubiquitin.

The region encompassing the CC2, UBAN, and LZ domains (CC2-LZ) of NEMO has been shown to be necessary and sufficient for di-ubiquitin binding, and missense mutants in the LZ (L329P) or the UBAN domain (D311N) disrupt binding to polyubiquitin chains⁹. Moreover, the CC2-LZ region of NEMO is indispensable for LUBAC binding^{10,11}. Both the UBAN domain

and ZF domain of NEMO are involved in ubiquitin binding¹². Whereas the UBAN domain preferentially binds M1-linked ubiquitin chains, the ZF domain has a higher affinity for K63-linked ubiquitin chains^{13, 14}.

Another important feature of NEMO is its dimerization/oligomerization, which is essential for binding to ubiquitin^{15, 16}. The LZ was reported to arrange the Cys347 residues in close proximity for disulfide bond formation in a NEMO homodimer¹⁷. In conclusion, mutations affecting NEMO ubiquitin binding and dimerization most probably impair the interaction between NEMO and p62 and thus protein aggregate clearance.

3. Figure 4A needs a quantification, not just images of 3 cells. Especially since in the 3rd image many cells do not show recruitment of NEMO to a-Syn-GFP.

The experiment has been performed with SH-SY5Y cells stably expressing GFP-a-synuclein that were transiently transfected with plasmids encoding wildtype NEMO or Q330X NEMO. This is why not all cells with synuclein-GFP aggregates (green) express NEMO (red), based on the low transient transfection efficiency. For example, in the third row, some cells next to the synuclein-positive cell do not express NEMO, therefore no NEMO recruitment to synuclein aggregates can be seen in these cells. We apologize for not adequately explaining the experimental approach in the previous version. In the revised version, we included a quantification of the experiment shown in Fig. 4A.

4. The authors use BafA1 treatment to block p62-mediated aggregate clearance through autophagosome-lysosome degradation, however further experiments are needed to fully prove this. For example: co-staining of aggregates with LC3B or LAMP1 as autophagosome-lysosome markers; in ATG5 or FIP200 KO cells.

We performed co-stainings of synuclein aggregates with LC3 and LAMP2 in wildtype and NEMO-deficient cells. The diffuse, non-condensated staining of p62 in NEMO-deficient cells was accompanied by a significantly reduced abundance of both LC3 and LAMP2 at synuclein aggregates (Fig. 7F, G). Moreover, we observed that also NBP1, which has been reported to co-condensate with p62¹⁸, is also impaired in forming foci at synuclein aggregates in the absence of NEMO (Fig. 7E).

We also made use of ATG5 KO mouse embryonic fibroblasts (MEFs) to test whether the effect of NEMO on the abundance of aSyn aggregates is impaired in the absence of ATG5. Although we tried hard, we were not able to obtain meaningful results, based on the fact that a transient aSyn seeding assays was required for this approach. Unfortunately, only a very small fraction of cells was targeted by both aSyn-GFP and untagged aSyn preformed fibrils and those cells were prone to cell death.

5. MG-132 treatments to show increased aggregates in NEMO KO cells should be compared to BafA1 treatment side-by-side.

As suggested, we treated NEMO KO cells in parallel with BafA1 and found that similarly to MG-132, BafA1 caused an increase in cells with protein aggregates the absence of NEMO (Fig. 2A, B).

Minor comments:

Introduction:

6. The role of p62 as an autophagy receptor for aggregates is well established and should already be referred to in the introduction. It might be beneficial for the reader to briefly explain the molecular mechanism of autophagy, in order to understand how p62 removes the protein aggregates.

We fully agree and included this important aspect already in the introduction of the revised version.

7. There should be more info on the fact that NEMO, ABINs and OPTN are highly similar in two aspects: 1) they harbor a UBAN domain and have all been identified as autophagy receptors.

This is indeed an interesting fact that we have taken up in the discussion of the revised version.

8. It should be explained what the LZ and ZF domains in NEMO are and what their functions include, since these domains are missing in the Q330X mutant and are potentially critical for the here discovered role of NEMO.

The NEMO CC2-LZ region includes the UBAN domain (Wagner et al. 2008) and forms elongated parallel coiled-coil dimers¹⁰. This region is required for binding to polyubiquitin, dimerization and interaction with HOIP^{10, 11, 19}. The UBAN domain binds M1-linked polyubiquitin chains 100-fold stronger compared to K63-linked chains, whereas the C-terminal ZF binds K63-linked polyubiquitin chains with high affinity^{12, 13, 14, 20, 21, 22}. The ZF has also been reported to mediate binding to I κ B α ²³ and to the ubiquitin chain-editing enzyme A20, a negative regulator of NF- κ B activation²⁴. We included this information in our revised manuscript.

Figure legends:

9. Figure legends for supplementary figures occur in the main text as well as in the supplementary figure file. One set should be removed.

We removed the legends for the supplementary figures from the main text.

Figures:

13 What is the cell viability in NEMO KO cells upon BafA1 treatment?

We tested the cell viability in wildtype and NEMO KO cells upon BafA1 treatment NEMO KO cells over time and observed that NEMO KO cells are more vulnerable to BafA1 treatment (without additional proteotoxic stress). In general, cell death in BafA1-treated cells occurred later than in MG-132-treated cells (Fig. S2A, B), suggesting that the impact of proteasomal inhibition on proteotoxicity is faster than that of lysosomal inhibition, at least in the cell type tested (mouse embryonic fibroblasts).

Reviewer #2:

In 'NEMO reshapes the protein aggregate interface and promotes aggregate clearance by co-condensation with p62', Furthmann et al. examine how NEMO promotes clearance of misfolded protein aggregates, e.g., α -synuclein aggregates in various cell culture model. It's interesting to note that they claim NEMO facilitates α -synuclein aggregate clearance by encouraging co-condensation with p62. This intriguing study sheds new, perhaps significant light on the process underlying aggregate clearance. The authors' intriguing and original work is noteworthy but before I can suggest publication, there are some issues that need to be addressed.

Major issues:

1. The authors claimed that misfolded α -synuclein is decorated with M1-linked ubiquitin and NEMO. In this manuscript, there is minimal proof that α -synuclein is M1 ubiquitinated or that it interacts with NEMO. Comparing the co-localization of NEMO and α -synuclein to that of LUBAC and α -synuclein, the immunofluorescence images in Fig. 3D are ambiguous. The author should evaluate the interaction between α -synuclein aggregates and NEMO.

In the revised manuscript we added new data and improved the presentation of our data to clarify these issues. Our previous presentation may have appeared ambiguous, since only a magnified α Syn aggregate was shown in the inset. In the revised presentation, also the α Syn-negative cytoplasm around the aggregate is shown. Thus, colocalization of α Syn aggregates with wildtype NEMO in contrast to mutant NEMO, enrichment of M1-ubiquitin chains and LUBAC components at α Syn aggregates is now obvious (Fig. 3A, B, D, E; Fig. 4A; Fig. 5A; Fig. S6A, C). Please note that in Fig. 3D endogenous NEMO is shown, which is difficult to visualize in immunocytochemistry. Yet, a significant increase in the fluorescence intensity of endogenous NEMO is seen at α Syn aggregates in comparison to the cytoplasm (Fig. 3D).

Moreover, we did not claim a direct interaction between α Syn and NEMO. The recruitment of NEMO to α Syn aggregates is dependent on polyubiquitin chains, in particular on M1-linked chains, and HOIP, since NEMO directly binds to M1-linked ubiquitin chains and HOIP. HOIP is recruited by a direct interaction of its PUB domain with the PIM domain of VCP/p97

²⁵, explaining why we found M1-linked ubiquitin chains at all intracellular protein aggregates we analyzed so far.

To add more biochemical evidence for the recruitment of NEMO to protein aggregates, we used cells expressing mutant Huntingtin (Htt-polyQ). In the aSyn seeding model, a-synuclein aggregates are challenging to analyze biochemically. Unlike other aggregation-prone proteins linked to neurodegeneration, aSyn does not form aggregates by increased expression in cellular models. To induce aSyn aggregation, cells stably expressing aSyn-GFP are treated with aggregated recombinant aSyn without GFP (preformed fibrils or “seeds”). Depending on the seeding efficiency, only a subpopulation of the cells takes up the preformed fibrils and shows aggregation of aSyn-GFP. Thus, the biochemical analysis is hampered by large amounts of soluble a-synuclein-GFP and the presence of untagged fibrils, which can create post-lysis artefacts. To illustrate this phenomenon, we performed a detergent solubility assay of cellular aSyn aggregates, showing the Triton X-100-soluble, Sarcosyl-soluble and Sarcosyl-insoluble fractions. We calculated that aSyn-GFP is increased by a factor of 210 in the Triton X-100-soluble fraction in comparison to the Sarcosyl-insoluble fraction (Fig. R1). However, wildtype NEMO in contrast to Q330X NEMO was enriched in the insoluble fraction containing aSyn-GFP aggregates (Fig. R1).

To address the reviewer’s concerns and add more biochemical data, we used cells expressing Huntingtin with an extended polyglutamine stretch (Htt-Q97), since overexpression of Htt-Q97 results in the formation of aggregates in nearly all cells. With the help of this model, we have shown before that NEMO is recruited to Htt-polyQ aggregates in a HOIP-dependent manner ²⁵ (EV4E), see Fig. R2. We performed filter retardation assays which allow detection of SDS-insoluble proteins on a cellulose acetate membrane, whereas SDS-soluble proteins are filtered through the membrane. In lysates from cells co-expressing Htt-Q97 together with either wildtype NEMO or Q330X NEMO or K285R/K309R NEMO (a mutant that is impaired in M1-ubiquitination and binding to M1-linked ubiquitin), wildtype NEMO was retained together with Htt-Q97, whereas retention of both Q330X NEMO and K285R/K309R NEMO was strongly reduced (Fig. 4C).

Fig. R1. Wildtype NEMO and aSyn-GFP co-partition into the Sarcosyl-insoluble fraction in aSyn-seeded SH-SY5Y cells. SH-SY5Y cells stably expressing aSyn A53T-GFP were transiently transfected with HA-tagged wildtype NEMO or Q330X NEMO or a vector control (pcDNA3.1). After 24 h, the cells were treated with recombinant aSyn A53T seeds and harvested 48 h after seeding. The cells were lysed in 0.5 % (v/v) Triton X-100 and 0.5 % (v/v) desoxycholate in PBS supplemented with NEM (Sigma), protease inhibitor (cOmplete, Roche) and phosphatase inhibitors (PhosStop, Roche). Samples were centrifuged (20 min, 20.000 x g, 4 °C), the supernatants were transferred into a new tube and 4 x Laemmli sample buffer was added (**Triton X-100-soluble fraction**). The pellets were lysed 1 % (v/v) Sarcosyl in 10 mM Tris supplemented with NEM (Sigma), protease inhibitor (cOmplete, Roche), phosphatase inhibitor (PhosStop, Roche), MgCl₂ (Serva) and DNase I (PanReac). The samples were centrifuged (20 min, 20.000 x g, 4°C), the supernatants were transferred into a new tube and 4 x Laemmli sample buffer was added (**Sarcosyl-soluble fraction**). 2 x Laemmli sample buffer was added to the pellets (**Sarcosyl-insoluble fraction**). 6% of the Triton X-100-soluble fraction, 12% of the Sarcosyl-soluble fraction and 100% of the Sarcosyl-insoluble fraction were loaded on SDS-PAGE and analyzed by immunoblotting using the indicated antibodies.

Fig. R2. HOIP is required to recruit NEMO to Htt-Q97. NEMO (red) and Htt-Q97-GFP (green) were analyzed by immunocytochemistry in wildtype HAP1 cells (control), HOIP-KO HAP1 cells, or HOIP-KO HAP1 cells reconstituted with HOIP. DAPI (blue). Scale bar, 10 μ m. Source: van Well et al., 2019 (EV4E)

2. The author should address NEMO-LUBAC dependent ubiquitination of a-synuclein. The IPs are far more convincing evidence.

As outlined above, we used cells expressing Htt-Q97 to address the role of NEMO in M1-ubiquitination of aggregated proteins by an unambiguous biochemical approach. It turned out that only wildtype NEMO but not Q330X NEMO or D311N NEMO, a mutant defective in binding to linear ubiquitin chains, can increase the abundance of M1-linked ubiquitin at Htt-polyQ aggregates (Fig. 4I). These results are in line with our data from quantitative image analysis, indicating that wildtype NEMO significantly increased M1-ubiquitination at synuclein aggregates (Fig. 4H).

3. The authors suggested that NEMO causes the co-condensation of a-synuclein aggregates with p62, creating a mobile phase transition that is necessary for effective autophagosomal clearance. It is essential to address the recruitment to LC3 and autophagosome components.

We fully agree and therefore visualized and quantified the recruitment of LC3 and LAMP2 to aSyn aggregates in wildtype and NEMO KO cells (Fig. 7F, G). In wildtype cells, the foci-like pattern of p62 staining at aSyn aggregates was also seen for the selective autophagy receptor NBR1 (Fig. 7E), which has been reported to co-condensate with p62¹⁸. Strikingly, the diffuse, non-condensated staining of p62 and NBR1 in NEMO-deficient cells was accompanied by a reduced abundance of LC3 and LAMP2 at aSyn aggregates, suggesting that the reduced focal enrichment of p62/NBR1 has functional consequences on the subsequent recruitment of the autophagic machinery.

4. Rudenko et al. demonstrated that Δ CT-NEMO stimulates NF- κ B signaling and inflammation. The authors need to discuss about how this manuscript differs from their previous work.

Zilberman-Rudenko and colleagues studied the interesting and seemingly paradox phenomenon that some NEMO mutations cause an autoinflammatory phenotype. They studied the E391X NEMO mutation that causes X-linked ectodermal dysplasia with anhidrosis and immunodeficiency combined with inflammatory disease phenotypes, such as colitis and dermatitis. By using primary CD4⁺ T cells from patients and reconstituted Jurkat cells, the authors of this study found that the E391X mutant shows increased NF- κ B activity compared to wildtype NEMO, both under basal conditions and in response to TNF or TLR ligands. It turned out that E391X NEMO is impaired in interacting with A20, a negative regulator of NF- κ B activation, based on its ubiquitin chain-editing function. A20 promotes the conversion of K63- to K48-ubiquitinated RIP1 in the TNFR complex, thereby inducing degradation of RIP1. Reduced recruitment of A20 by E391X NEMO preserves K63-ubiquitinated RIP1 and enhances IKK activation.

In contrast to the Q330X NEMO mutant, the E391X mutant can obviously still interact with M1-/K63-linked ubiquitin chains via the UBAN domain, explaining its ability to promote NF- κ B activation. In addition, cell type-specific effects may play a role, since defective IKK activation has been reported in various cell types expressing NEMO lacking the C-terminal ZF domain ²⁶.

Minor issues:

1. In Figures 1C and D, the authors omitted the control brain.

We performed immunohistochemistry also with samples from control brain (Fig. R3). If the Reviewer feels that we should include this information as a supplementary figure, we will be happy to do so.

Fig. R3. Immunohistochemistry of NEMO, α -synuclein, Tau, and TDP-43 in human control brain tissue. Immunofluorescent stainings of cortical (Tau, TDP-43) or midbrain (α -synuclein) sections from control brains corresponding to Fig. 1C, D. Brain sections were stained with antibodies against NEMO, and α -synuclein, Tau, or TDP-43; DAPI (blue). Scale bar, 10 μ m.

2. In contrast to other samples, the NEMO fluorescence signal in the PD sample in Figure 1D appears to be stronger. Is the NEMO expression higher in the PD sample?

This is an interesting question, which we tried to address by imaging a larger number of brain sections from PD, AD and FTD patients. However, we got the impression that too many variables may influence the intensity of NEMO staining, such as the brain region analyzed, the age of the patients and *post mortem* interval. Therefore, we do not want to overinterpret our data.

3. Fig. 1D's depiction of NEMO and α -synuclein colocalization is unclear. In comparison with Tau and TDP-43, NEMO showed only limited colocalization with α -syn in PD.

In the revised version we show a larger magnification of the image, showing the foci-like enrichment of NEMO at the α -synuclein aggregate (new Fig. 1D).

4. Images of NEMO KO MEF in Fig. 2A cannot show the differences in the percentage of cells with aggregates.

In the revised version, we included BafA1 treatment of wildtype and NEMO KO cells as another stress condition, as suggested by Reviewer #1. We showed a larger number of cells per image and presented the results in form of a bar graph, as the differences between wildtype and NEMO KO cells are difficult to see when looking only at a section of the coverslips (Fig. 2A, B). The data are based on the evaluation of at least 150 cells per condition.

5. Fig. 2C and D are missing every single data point.

In the revised version, we showed the single data points in Fig. 2C and D.

6. In Fig. 2F, TNF caused the activation of NF- κ B whereas heat shock to WT HEK293 cells did not. Does NEMO phase separation or ubiquitination alter under these conditions?

Yes, NEMO forms M1-positive assemblies in heat stressed cells. This is part of another project we are currently following up.

7. As previously indicated, the colocalization of a-synuclein and NEMO is still unclear in Fig. 3D. The signals of NEMO were blurry.

In the revised version, we optimized the presentation of the images and showed also the cytoplasm surrounding the aSyn aggregates so that the enrichment of endogenous NEMO at aSyn aggregates is obvious. In addition, we quantified the intensities of the NEMO-specific fluorescent signals in the cytoplasm and at aSyn aggregates (Fig. 3D).

8. Colocalization analysis is necessary in Fig. 4A.

Due to the fact that NEMO has been transiently transfected into cells stably expressing synuclein-GFP, only a subfraction of aSyn-expressing cells are positive for HA-NEMO expression. For example, in the third row, some cells next to the aSyn-positive cell do not express NEMO, therefore no NEMO recruitment to synuclein aggregates can be seen in these cells. In the revised version, we included a quantification of NEMO colocalizing with aSyn aggregates (Fig. 4A).

9. No fluorescence images for Fig. 4F and G. For evidence of M1 ubiquitination of a-synuclein aggregates in Fig. 4, IP experiments are required.

As outlined above, the biochemical analysis of synuclein aggregates in the seeding model is challenging. We therefore used the Htt-polyQ cellular model to add more evidence for a NEMO-dependent increase in M1-ubiquitination at aggregates (Fig. 4I).

Fluorescence images corresponding to Fig. 4F and G (new Fig. 4G and H) are added as a supplementary figure (Fig. S6A).

10. In Fig. 5A, P65 was enriched at a-synuclein aggregates following a-synuclein seeding, whereas in Fig. 2F, P65 was dispersed following heat shock. The potential differences between these two scenarios should be discussed by the authors.

The intention of the experiments shown in Fig. 2E and 2F was to test whether a transient heat stress induces NF- κ B activation. We found no evidence for NF- κ B-mediated

transcriptional activity (Fig. 2E) or p65 translocation (Fig. 2F), suggesting that a transient heat stress is not an NF- κ B-activating stimulus. During transient heat stress, there is usually no formation of irreversible aggregates, where p65 could be sequestered.

Reviewer #3:

NEMO is conventionally known and described as a scaffold protein in the NF- κ B transcription factor pathway, where NEMO assembles the kinases (and other proteins) necessary for activation of the pathway. In addition, there are human immunodeficiency diseases that are caused by mutations in the X-linked NEMO gene, and these mutations have generally been described as reducing NEMO's function in NF- κ B signaling. As such, many of NEMO's functions have been described in immune cells.

In the paper by Furthmann et al they have identified what appears to be a new NEMO pathway/function, which is independent of NF- κ B, wherein NEMO functions in a proteostasis pathway. In this pathway, NEMO is important for protein aggregate/misfolded protein clearance. Moreover, at least one patient with a NEMO mutation has a widespread brain protein aggregates that involves defective p62 and phase separation. That is, the authors suggest that an additional normal function of NEMO is to promote autophagosomal clearance of protein aggregates by promoting condensation of the liquid-liquid phase separation p62 protein condensates. Thus, these results indicate that NEMO may have a function in neuronal cells and that in certain cases human mutations can also lead to neuronal disease in humans. Overall, this is an extensive series of results with somewhat interesting implications at least in the patient under study.

Concerns

---How good is the evidence that the only mutation in these people that lead to the neuropathy is a NEMO mutation?

During the revision of this manuscript, we performed whole-genome sequencing (WGS) of the Q330X NEMO patient to check whether other mutations could have caused the progressive neurodegenerative phenotype. WGS did not identify pathogenic or likely pathogenic variants in genes associated with neurodegenerative disorders (Suppl Fig. S1). Notably, the patient did not have pathogenic mutations in *p62/SQSTM1*, *OPTN*, or *TBK1*. In addition, the Q330X NEMO mutation occurred *de novo* in our patient (confirmed by mosaicism found in the WGS) and there was no family history of neurodegenerative diseases. Therefore, all evidence supports the notion that the phenotype of this patient was caused by the NEMO variant. A causal relationship is further supported by our characterization of the Q330X NEMO mutant *in vitro* and in cellular models in combination with the increased vulnerability of NEMO-deficient cellular models to proteostasis dysregulation, supporting a role of NEMO in preventing the accumulation of protein aggregates in general. This is also reflected by the neuropathological alterations seen in the patient's brain.

---It is unclear to me at what level NEMO protein is expressed in neurons. Is that shown to some extent in Fig. 1C/D? Could the authors investigate the NEMO protein/mRNA expression datasets of NEMO across tissues, esp as in brain compared to immune cells. In particular what does the more neuronal optineurin NEMO-like protein look like in this patient? I believe that OPTN mutations are more common in neuronal protein accumulation diseases than NEMO mutations, and given that some functions/domains of NEMO and OPTN overlap, there is some possibility that some of the findings in this paper are due to OPTN deficiencies that can be, at least partially, substituted by overexpression of NEMO.

As indicated by immunohistochemistry of control and patients' brain samples, NEMO is expressed in neurons. As suggested by this reviewer, we compared the expression levels of NEMO in different cells, including primary cortical neurons (PNs), induced peripheral neurons differentiated from pluripotent stem cells (iPNs), mouse embryonic fibroblasts (MEFs), and peripheral blood mononuclear cells (PBMCs) (Fig. R4).

We did not observe obvious differences in the expression of Optineurin in the Q330X NEMO patient's brain in immunohistochemistry. Notably, the patient did not have pathogenic mutations in *OPTN*.

Fig. R4. NEMO is expressed in neuronal cells. Primary cortical neuron (PNs, DIV7), induced peripheral neurons (iPNs, DIV21), mouse embryonic fibroblasts (MEFs) and peripheral mononuclear blood cells (PBMCs) were lysed in 1% Triton X-100 lysis buffer supplemented with protease inhibitors. 30 µg of protein was immunoblotted using antibodies against NEMO and β-actin.

--Fig. 1C and D. A few concerns

---I am guessing that the blue staining is DAPI or Hoechst but this is never really described, and is present in some panels but not others (?)

Yes, the blue staining is DAPI. We added this information to the figure legends.

---The co-localization is a bit tenuous. First, wouldn't any cytoplasmic protein "co-localize" with the cytoplasmic NEMO staining? Second, they show single cells in each case; were these selected in some way?

We are sorry for not clearly explaining our images. We do not show colocalization of two cytosolic proteins but enrichment of endogenous NEMO at α-synuclein aggregates in the cytoplasm, both in the brain and cellular models. In the revised manuscript we improved the

presentation of our data. We presented a higher magnification of the α -synuclein aggregate in the brain (Fig. 1D) and showed the cytoplasm surrounding the aggregates in immunocytochemistry. In addition, we included quantifications for the colocalization of endogenous NEMO and aSyn aggregates (Fig. 3D) and wildtype NEMO versus Q330X NEMO and aSyn aggregates (Fig. 4A).

---Fig. 2B, the differences in MG132 treated bars, based on error bars, do not seem especially significant (due to overlap of error bars).

---Fig. 2B, the first two bars are labelled as ns, but it appears that the untreated samples are about 5-10 fold different, which is greater than any of the other pairs. Should the authors simply normalize each same to the untreated control as 1.0, and how would the data look?

In the revised manuscript, we analyzed and quantified three new biological replicates, allowing us to also include BafA1 treatment, as suggested by Reviewer #1. We compared untreated cells with heat stressed, MG-132-treated, and BafA1-treated cells. The statistical analysis was performed after normalizing the untreated group as 0, and using two-way ANOVA followed by Šídák's multiple comparison test (Fig. 2A, B). Although upon MG-132 treatment there was a trend towards an increase in Proteostat-positive aggregates in NEMO deficient cells, the results are statistically non-significant in contrast to heat stress and BafA1 treatment.

---Fig. 2D, the way the error lines above bar graphs are designated do not in any way make it clear what is ns and what is significant. The authors should probably have bracket types of bars to show ns and/or significance. For example, when one looks at the ns bar that is above the first three bars in the graph, one is inclined to believe that the first three bars are not significantly different, but I think they mean that bars 1 and 3 are ns.

In the revised version, we concentrated on the most important effects, making the Fig. 2C and D more reader-friendly.

---Fig. 3C, the authors do not really describe which scale bars are 5 μ m and which are 20 μ m.

We included this information in the revised manuscript.

---Fig. 3D, again the overlap with NEMO is not especially convincing, e.g., that it is actually more than what one would see for any cytoplasmic protein

As outlined above, we improved the presentation of the images. The green staining (aSyn-GFP) shows a aSyn aggregate in the cytoplasm (black background). The red staining shows the enrichment of endogenous NEMO at the aggregates over the cytoplasm. In the revised

manuscript we quantified the fluorescence intensities of NEMO in the cytoplasm and at the aSyn aggregates (Fig. 3D).

----Fig 4A is potentially an important expt if it shows that WT but not mutant NEMO goes to aggregates. I am guessing the transfection efficiency is not 100% in these experiments. So how does one know that the cells expressing the Syn aggregates are indeed overexpressing the NEMO mutant. I am guessing that is from the first merge low power panels? What do WT NEMO vs mutant NEMO look like in a cell that does not have a Syn aggregate?

The experiment has been performed with SH-SY5Y cells stably expressing aSyn-GFP that were transiently transfected with plasmids encoding either wildtype HA-NEMO or Q330X HA-NEMO. This is why only a subfraction of the cells with aSyn-GFP aggregates (green) express NEMO (red, detected by an antibody against the HA tag), based on the low transient transfection efficiency. For example, in the third row, some cells next to the synuclein-positive cell do not express NEMO, therefore no NEMO recruitment to synuclein aggregates can be seen in these cells. In the revised version, we included a quantification corresponding to Fig. 4A.

In cells without aSyn aggregates, wildtype NEMO shows a diffuse staining, similarly to Q330X NEMO that is not recruited to synuclein aggregates.

---In Fig. 4B, the co-localization of NEMO with the aggregates seems to be around the aggregates. Is that what I am supposed to see?

Yes, the staining pattern depends on the compactness of the aggregates. Htt-polyQ aggregates are usually quite dense, so that the NEMO-specific antibodies cannot reach the core of the aggregates. Please note that Htt-Q97-GFP was used for this experiment, explaining why the green signal is seen all over the aggregates.

---The authors show that p62 coats Syn aggregates more robustly in NEMO KO as compared to NEMO WT cells (Fig. 8). Is that what one would expect based on their results? Is it consistent with this sentence: "Our study identified NEMO as a major player in priming the aggregate interphase for p62 condensation."

In the absence of NEMO, p62 shows a more diffuse staining in contrast to the foci-like staining in the presence of NEMO (Fig. 7D). The same holds true for the selective autophagy receptor NBR1 (Fig. 7E). This foci-like staining represents the condensed state of p62, which is a prerequisite to efficiently recruit the autophagic machinery. In the revised manuscript we added evidence for this: In the absence of NEMO, the diffuse pattern of p62 and NBR1 at synuclein aggregates goes along with a decrease in LC3 and LAMP2 recruitment to the aggregates (Fig. 7F, G).

References

1. Cammarata-Scalisi F, Fusco F, Ursini MV. Incontinentia Pigmenti. *Actas Dermosifiliogr (Engl Ed)* **110**, 273-278 (2019).
2. Conte MI, *et al.* Insight into IKBKG/NEMO locus: report of new mutations and complex genomic rearrangements leading to incontinentia pigmenti disease. *Hum Mutat* **35**, 165-177 (2014).
3. Senegas A, Gautheron J, Maurin AG, Courtois G. IKK-related genetic diseases: probing NF-kappaB functions in humans and other matters. *Cell Mol Life Sci* **72**, 1275-1287 (2015).
4. Fryssira H, Kakourou T, Valari M, Stefanaki K, Amenta S, Kanavakis E. Incontinentia pigmenti revisited. A novel nonsense mutation of the IKBKG gene. *Acta Paediatr* **100**, 128-133 (2011).
5. Fusco F, *et al.* Molecular analysis of the genetic defect in a large cohort of IP patients and identification of novel NEMO mutations interfering with NF-kappaB activation. *Hum Mol Genet* **13**, 1763-1773 (2004).
6. Aradhya S, *et al.* Multiple pathogenic and benign genomic rearrangements occur at a 35 kb duplication involving the NEMO and LAGE2 genes. *Hum Mol Genet* **10**, 2557-2567 (2001).
7. Sun S, *et al.* A novel inhibitor of nuclear factor kappa-B kinase subunit gamma mutation identified in an incontinentia pigmenti patient with syndromic tooth agenesis. *Arch Oral Biol* **101**, 100-107 (2019).
8. Fusco F, *et al.* Alterations of the IKBKG locus and diseases: an update and a report of 13 novel mutations. *Hum Mutat* **29**, 595-604 (2008).
9. Wu CJ, Conze DB, Li T, Srinivasula SM, Ashwell JD. Sensing of Lys 63-linked polyubiquitination by NEMO is a key event in NF-kappaB activation [corrected]. *Nat Cell Biol* **8**, 398-406 (2006).
10. Rahighi S, Iyer M, Oveisi H, Nasser S, Duong V. Structural basis for the simultaneous recognition of NEMO and acceptor ubiquitin by the HOIP NZF1 domain. *Sci Rep* **12**, 12241 (2022).
11. Tokunaga F, *et al.* Involvement of linear polyubiquitylation of NEMO in NF-kappaB activation. *Nat Cell Biol* **11**, 123-132 (2009).
12. Cordier F, Grubisha O, Traincard F, Veron M, Delepierre M, Agou F. The zinc finger of NEMO is a functional ubiquitin-binding domain. *J Biol Chem* **284**, 2902-2907 (2009).
13. Laplantine E, *et al.* NEMO specifically recognizes K63-linked poly-ubiquitin chains through a new bipartite ubiquitin-binding domain. *EMBO J* **28**, 2885-2895 (2009).
14. Rahighi S, *et al.* Specific recognition of linear ubiquitin chains by NEMO is important for NF-kappaB activation. *Cell* **136**, 1098-1109 (2009).
15. Agou F, *et al.* The trimerization domain of NEMO is composed of the interacting C-terminal CC2 and LZ coiled-coil subdomains. *J Biol Chem* **279**, 27861-27869 (2004).
16. Tegethoff S, Behlke J, Scheidereit C. Tetrameric oligomerization of IkappaB kinase gamma (IKKgamma) is obligatory for IKK complex activity and NF-kappaB activation. *Mol Cell Biol* **23**, 2029-2041 (2003).
17. Herscovitch M, *et al.* Intermolecular disulfide bond formation in the NEMO dimer requires Cys54 and Cys347. *Biochem Biophys Res Commun* **367**, 103-108 (2008).
18. Turco E, *et al.* Reconstitution defines the roles of p62, NBR1 and TAX1BP1 in ubiquitin condensate formation and autophagy initiation. *Nat Commun* **12**, 5212 (2021).
19. Fujita H, *et al.* Mechanism underlying IkappaB kinase activation mediated by the linear ubiquitin chain assembly complex. *Mol Cell Biol* **34**, 1322-1335 (2014).

20. Komander D, Reyes-Turcu F, Licchesi JD, Odenwaelder P, Wilkinson KD, Barford D. Molecular discrimination of structurally equivalent Lys 63-linked and linear polyubiquitin chains. *EMBO Rep* **10**, 466-473 (2009).
21. Wagner S, *et al.* Ubiquitin binding mediates the NF-kappaB inhibitory potential of ABIN proteins. *Oncogene* **27**, 3739-3745 (2008).
22. Lo YC, *et al.* Structural basis for recognition of diubiquitins by NEMO. *Mol Cell* **33**, 602-615 (2009).
23. Schrofelbauer B, Polley S, Behar M, Ghosh G, Hoffmann A. NEMO ensures signaling specificity of the pleiotropic IKKbeta by directing its kinase activity toward IkappaBalpha. *Mol Cell* **47**, 111-121 (2012).
24. Zilberman-Rudenko J, *et al.* Recruitment of A20 by the C-terminal domain of NEMO suppresses NF-kappaB activation and autoinflammatory disease. *Proc Natl Acad Sci U S A* **113**, 1612-1617 (2016).
25. van Well EM, *et al.* A protein quality control pathway regulated by linear ubiquitination. *EMBO J* **38**, (2019).
26. Du M, Ea CK, Fang Y, Chen ZJ. Liquid phase separation of NEMO induced by polyubiquitin chains activates NF-kappaB. *Mol Cell*, (2022).

Reviewer #1 (Remarks to the Author):

The authors have done excellent work in responding to all raised questions. The manuscript is now acceptable for publication

Ivan Dikic

Reviewer #2 (Remarks to the Author):

The authors did an extra work to address the reviewers. I believe that the paper is much improved. but I do have 1 minor comment that they should address.

Figures 4C and R1 show a similar level of Q97-GFP across all groups. In line with these, NEMO Q330X or D311N-expressing cells displayed comparable levels of Q97-GFP in the insoluble (formic acid soluble) fraction as compared to WT-expressing cells. It is unclear if the solubility and accumulation of those proteins is affected directly by loss-of-M1 ubiquitylation.

Reviewer #3 (Remarks to the Author):

In the revised paper by Furthmann et al they have characterized what appears to be a new NEMO pathway/function, which is independent of its normal NF- κ B pathway function. In this new pathway, NEMO functions in a proteostasis pathway. That is, NEMO is important for protein aggregate/misfolded protein clearance. Moreover, at least one patient with a NEMO mutation has widespread brain protein aggregates that involve defective p62 and liquid-liquid phase separation. Thus, the authors suggest that an additional function of NEMO is for autophagosomal clearance of protein aggregates by promoting condensation of the liquid-liquid phase separation p62 protein condensates. Overall, these results indicate that NEMO may have a function in neuronal cells and that in certain cases mutations can also lead to neuronal disease in humans. This is an extensive series of results with interesting implications at least in the patient under study.

The authors have improved the paper by including additional data and analyzing previous data in greater detail. In particular, the whole genome sequence addresses the major concern asking for proof that the patient's disease is not due to a protein/gene other than NEMO.

A few minor comments

--Fig. 4C, why does mutant K285R/K309R migrate higher than WT NEMO?

--Fig. 4E, I am unclear on the cropping of the top panel. I don't think that the panel would include the approximately 40 kDa Q330X mutant even if it were there (?)

--Line 213, at end of sentence add "to aSyn aggregates"

--Line 440, add kappa symbol font to NF- κ B

NCOMMS-23-05887B

Rebuttal Letter

Reviewer #1:

We are happy to hear that this reviewer is pleased with our revised manuscript.

Reviewer #2:

The authors did an extra work to address the reviewers. I believe that the paper is much improved, but I do have 1 minor comment that they should address.

Figures 4C and 4I show a similar level of Q97-GFP across all groups. In line with these, NEMO Q330X or D311N-expressing cells displayed comparable levels of Q97-GFP in the insoluble (formic acid soluble) fraction as compared to WT-expressing cells. It is unclear if the solubility and accumulation of those proteins is affected directly by loss-of-M1 ubiquitylation.

The experiments shown in Fig. 4c and 4i were conceived to show the association of NEMO (4c) or M1-linked ubiquitin (4i) with Htt-Q97. However, these approaches are not suitable to detect influences of NEMO or M1-linked ubiquitylation on the solubility or accumulation of Htt-polyQ variants. The Htt-Q97-GFP construct used for these experiments drives the expression of a highly aggregation-prone Htt variant with rapid aggregation kinetics under control of a strong CMV promoter.

This is why we are currently addressing these aspects by using different approaches. First, we are using Htt-Q40 and Htt-Q60, which show slower aggregation kinetics. Second, we are studying aggregation and the clearance of aggregates by pulse/chase experiments, such as optical pulse labeling with photo-convertible proteins (Dendra2-Htt-polyQ constructs) and cycloheximide chase experiments in inducible cell lines.

On a side note, the increased abundance in Htt-Q97-only expressing cells in Fig. 4i may be caused by the fact that vector DNA was used to equal the amount of transfected plasmid DNA. Thus, there was no competition for transcription and translation in the Htt-Q97-only sample in comparison to the samples containing Htt-Q97 and NEMO plasmids.

Reviewer #3:

In the revised paper by Furthmann et al they have characterized what appears to be a new NEMO pathway/function, which is independent of its normal NF- κ B pathway function. In this new pathway, NEMO functions in a proteostasis pathway. That is, NEMO is important for

protein aggregate/misfolded protein clearance. Moreover, at least one patient with a NEMO mutation has widespread brain protein aggregates that involve defective p62 and liquid-liquid phase separation. Thus, the authors suggest that an additional function of NEMO is for autophagosomal clearance of protein aggregates by promoting condensation of the liquid-liquid phase separation p62 protein condensates. Overall, these results indicate that NEMO may have a function in neuronal cells and that in certain cases mutations can also lead to neuronal disease in humans. This is an extensive series of results with interesting implications at least in the patient under study.

The authors have improved the paper by including additional data and analyzing previous data in greater detail. In particular, the whole genome sequence addresses the major concern asking for proof that the patient's disease is not due to a protein/gene other than NEMO.

A few minor comments

--Fig. 4C, why does mutant K285R/K309R migrate higher than WT NEMO?

We assume that the difference in the SDS-PAGE migration pattern of NEMO K285R/K309R observed in Fig. 4c is caused by a "smile effect", since we did not observe differences in the migration on other immunoblots. For example, see the immunoblot shown below.

HEK293T cells were transiently transfected with plasmids encoding the indicated NEMO plasmids (MT1: D311N; MT2: A310E; UT: untransfected). Cells were lysed under denaturing conditions and analyzed by immunoblotting using an antibody specific for NEMO.

--Fig. 4E, I am unclear on the cropping of the top panel. I don't think that the panel would include the approximately 40 kDa Q330X mutant even if it were there (?)

A very faint band at approximately 40 kDa is present in the cropped top panel, which is difficult to see in the PDF version. For clarification, see the source data corresponding to Fig. 4e of our manuscript. In the revised version, we included a larger section of the blot.

--Line 213, at end of sentence add "to aSyn aggregates"

--Line 440, add kappa symbol font to NF-kB

Thank you for pointing this out. We addressed these issues in the revised version.